# FINE-GRAINED AUDIO-VISUAL JOINT REPRESENTATIONS FOR MULTIMODAL LARGE LANGUAGE MODELS

## ABSTRACT

Audio-visual large language models (LLM) have drawn significant attention, yet the fine-grained combination of both input streams is rather under-explored, which is challenging but necessary for LLMs to understand general video inputs. To this end, a fine-grained audio-visual joint representation (FAVOR) learning framework for multimodal LLMs is proposed in this paper, which extends a text-based LLM to simultaneously perceive speech and audio events in the audio input stream and images or videos in the visual input stream, at the frame level. To fuse the audio and visual feature streams into joint representations and to align the joint space with the LLM input embedding space, we propose a causal Q-Former structure with a causal attention module to enhance the capture of causal relations of the audio-visual frames across time. An audio-visual evaluation benchmark (AVEB) is also introducedproposed which comprises six representative single-modal tasks with five cross-modal tasks reflecting audio-visual co-reasoning abilities. While achieving competitive single-modal performance on audio, speech and image tasks in AVEB, FAVOR achieved over 20% accuracy improvements on the video question-answering task when fine-grained information or temporal causal reasoning is required. FAVOR, in addition, demonstrated remarkable video comprehension and reasoning abilities on tasks that are unprecedented by other multimodal LLMs. An interactive demo of FAVOR is available at `https://github.com/BriansIDP/AudioVisualLLM.git`, and the training code and model checkpoints will be released soon.

## 1 INTRODUCTION

Text-based large language models (LLM) (Brown et al., 2020; Touvron et al., 2023; Chiang et al., 2023; Anil et al., 2023; Du et al., 2022) have demonstrated remarkable performance in various natural language processing tasks, especially achieving human-level capabilities in reasoning and comprehension (OpenAI, 2023). Meanwhile, instruction fine-tuning (Chung et al., 2022; Ouyang et al., 2022; Peng et al., 2023), where data is organised as pairs of user instruction (or prompt) and reference response, has emerged as a training paradigm that enables LLMs to perform various tasks by following open-ended natural language instructions from non-expert users.

Recently, there has been a burgeoning research interest in equipping LLMs with visual and auditory perception abilities. While most recent studies have been focusing on incorporating a single specific type of input, such as image (Li et al., 2023a; Alayrac et al., 2022; Dai et al., 2023), video (Maaz et al., 2023; Chen et al., 2023b; Zhao et al., 2022; Zeng et al., 2023), audio (Gong et al., 2023) or speech (Zhang et al., 2023a; Rubenstein et al., 2023) separately. These investigations often employ a trained modality alignment module that aligns the representation space of the input modality with the text one. Subsequently, work has started looking at incorporating multiple simultaneous input modalities (Su et al., 2023; Zhang et al., 2023b; Lyu et al., 2023; Zhao et al., 2023; Chen et al., 2023a). Despite the sequential nature of video and audio inputs, most aforementioned work treated video as a sampled subset of individual images and audio as a fixed-length spectrogram. As a result, these models tend to ignore information and causal relations when the input sequence length increases. Moreover, speech, as a crucial aspect of auditory input in videos that in particular relies on fine-grained information extraction, is considerably under-explored in multimodal LLM research.

To this end, this paper proposes FAVOR, a **f**ine-grained **a**udio-**v**isual j**o**int **r**epresentation learning framework for LLM-based multimodal understanding and reasoning with audio-visual input

sequences consisting of images, audio events, speech, and video. It takes audio-visual sequences at high temporal resolution ~~certain frame rates~~ as inputs and, if paired, temporally synchronises them using a synchronisation module. Such a frame-level fine-grained synchronisation allows a more thorough and fine-grained interaction between audio and visual modalities across time, which is particularly beneficial for videos with speech. Since the input sequences have variable lengths, FAVOR divides the sequence into a number of fixed-length sliding windows and aligns the synchronised sequence within each window to the LLM input text representation space. In order to capture the causal relations among consecutive video frames within a window, a causal Q-Former structure is proposed that introduces a causal attention module to Q-Former (Li et al., 2023a).

FAVOR is comprehensively evaluated using an audio-visual evaluation benchmark (AVEB) introduced~~proposed~~ in this paper, which integrates 11 tasks including 6 different types of open-source tasks with single-modal inputs, as well as 5 cross-modal inference tasks. While achieving competitive performance on single-modal tasks, FAVOR also achieved large performance improvements on cross-modal tasks compared to single-modal models, e.g. over 10% absolute accuracy improvement on audio-visual sound source detection. Notably, benefiting from the fine-grained nature, FAVOR achieved a remarkably 25% accuracy improvement in video QA tasks compared to the strong InstructBLIP baseline. The main contribution of this paper can be summarised as follows:

- This paper proposes the FAVOR learning framework for multimodal LLMs. To the best of our knowledge, FAVOR is the first approach that is capable of performing cross-modal cognitive tasks involving audio, speech, image and video inputs with high temporal resolution.
- This paper proposes the causal Q-Former structure which comprises a causal encoder module. A novel diversity loss is also proposed to encourage diverse joint representations to be learned. ~~Further with a novel diversity training loss, causal Q-Former is capable of handling audio-visual sequence input efficiently with a small number of training examples.~~
- This paper introduces the AVEB benchmark comprising single-modal and cross-modal tasks to quantitatively evaluate the performance of audio-visual LLMs.

## 2 RELATED WORK

Our work is based on the Q-Former structure to fuse the audio and visual modalities and to align with the text representation space (Li et al., 2023a; Dai et al., 2023). While Q-Former has been primarily proposed for visual information extraction, it also performs remarkably in extracting auditory features for automatic speech recognition (ASR) (Yu et al., 2023). In addition to Q-Former, various types of modality aligners have been studied, such as the cross-attention mechanism (Alayrac et al., 2022), pre-trained multimodal embeddings, (Girdhar et al., 2023) and temporal and spatial pooling (Maaz et al., 2023). Different from standard Q-Former approaches, our causal Q-Former used in the FAVOR framework pays particular attention to the sequential nature of the input feature streams with the model structure and training methods dedicated to audio-visual understanding.

The work most closely related to ours is Video-LLaMA (Zhang et al., 2023b), Macaw-LLM (Lyu et al., 2023) and X-LLM (Chen et al., 2023a), as all of them used LLMs for cross-modal understanding based on general non-silent video inputs (referred to as audio-visual sequence in this paper). X-LLM supports video and Chinese speech inputs, but cannot understand audio events and music. Video-LLaMA employs an additional video Q-Former to encode features of several equally-spaced frames extracted using a BLIP2 (Li et al., 2023a) image encoder. Macaw-LLM adopted a similar approach and used three separate encoders for image, video and non-speech audio events. Both Video-LLaMA and Macaw-LLM consider only non-speech audio events, and the audio encoders in the two models are the ImageBind (Girdhar et al., 2023) and Whisper (Radford et al., 2023) model encoders respectively. While both methods involve the fusion of audio and visual feature streams, the two streams are sparsely pooled and processed rather independently, which removes fine-grained audio-visual interactions at each time step. Compared to Video-LLaMA and Macaw-LLM, FAVOR reserves fine-grained modality interactions and can understand speech inputs that are common in general non-silent videos. This leads to an emphasis on causal modality synchronisation across time and allows more content-based cross-modal interactions.

## 3 METHODOLOGY

In this section, we present the proposed FAVOR learning framework, which is designed to handle audio and visual input sequences synchronously at high temporal resolution for LLMs. This section

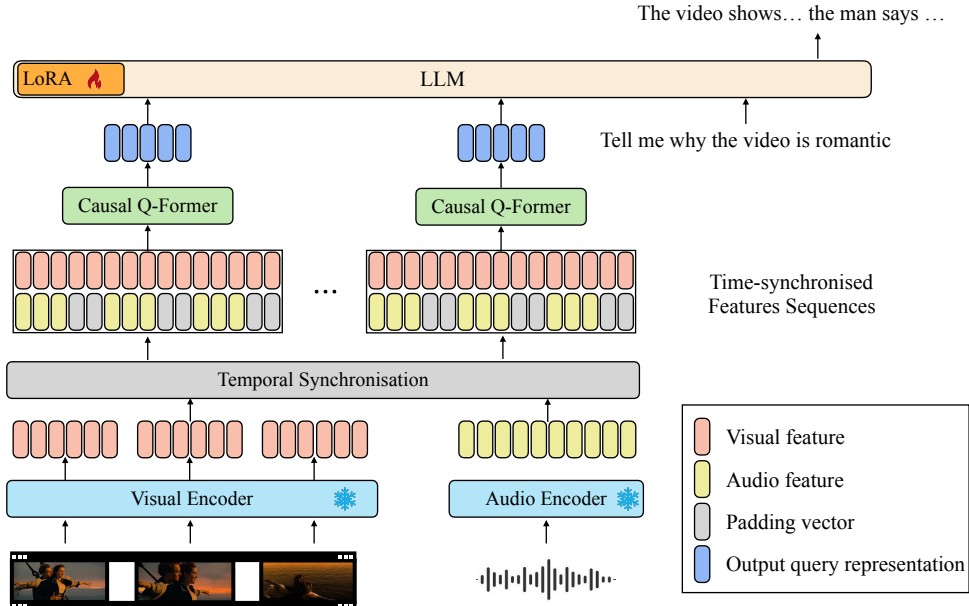

Figure 1: The fine-grained audio-visual joint representation (FAVOR) learning framework for multi-modal LLMs. The temporal synchronisation module does not contain trainable parameters, and the audio and visual feature encoders are not updated during training.

introduces the model structure, including the causal attention module and an optional diversity loss.

## 3.1 MODEL ARCHITECTURE

The structure of FAVOR is shown in Fig. 1. Key components that realise the fine-grained audio-visual representation learning are the temporal synchronisation module and the causal Q-Former. First, visual and audio inputs are encoded using the corresponding pre-trained encoders. The visual encoder in FAVOR converts the input image into a certain number of vectors via the image encoder in InstructBLIP (Li et al., 2023a). When video input is given, the visual encoder encodes each video frame separately as a sequence of images at a 2 Hz frame rate, and the output image features are concatenated along the temporal dimension to form a sequence of visual frames. The audio encoder used is the Whisper ASR model encoder (Radford et al., 2023) that converts the input speech and audio events into a sequence of vectors at a 50 Hz frame rate.

When both audio and visual inputs are present, the two encoded feature sequences are sent to the temporal synchronisation module to obtain the time-synchronised feature sequences, as shown in Fig. 1. Since video is sampled at a lower frame rate than audio, the audio and visual frames are synchronised at each video frame (*i.e.* every 0.5 seconds), with zero padding to make both sequences have equal lengths. Note that higher frequencies of visual frames are also supported in the FAVOR framework which requires higher computation and storage costs. The synchronised audio frame $\mathbf{h}_t^{\mathrm{A}}$ and visual frame $\mathbf{h}_t^{\mathrm{V}}$ are then concatenated along the feature dimension to obtain the combined audio-visual feature frame $\mathbf{h}_t^{\mathrm{AV}}$. That is,

$$\mathbf{h}_t^{\mathrm{AV}} = \mathrm{Concat}(\mathbf{h}_t^{\mathrm{A}}, \mathbf{h}_t^{\mathrm{V}}), \tag{1}$$

where $\mathrm{Concat}(\cdot)$ represents the concatenation along the feature dimension. Note that in cases when only one input modality is present, the other modality is filled with a sequence of zero padding of the same sequence length. While an image alone is treated as a single frame, when paired audio input exists, such as images with spoken captions (Hsu et al., 2020), each image is duplicated as if it were a video input with a matched length to the audio input.

In order to handle variable-length inputs, the combined feature sequences are first divided into fixed-length windows spanning, *e.g.* every 5 or 10 seconds. Then, a causal Q-Former based on the same $N$ trainable input query tokens $\mathbf{q}_1, \ldots, \mathbf{q}_N$ is applied to convert each sliding window and generate

$N$ output query vectors carrying the audio-visual information. As shown in Eqn. (2),

$$\mathbf{h}_{w,1}^{Q}, ..., \mathbf{h}_{w,N}^{Q} = \text{Q-Former}_{\text{causal}}(\mathbf{h}_{t}^{AV}, \ldots, \mathbf{h}_{t+k}^{AV}; \mathbf{q}_1, \ldots, \mathbf{q}_N), \quad (2)$$

where $w$ is the window index and $k$ is the number of video frames in that window, and Q-Former$_{\text{causal}}(\cdot)$ denotes the causal Q-Former computation described in detail later in Section 3.2. The output query representations, $\mathbf{h}_{w,1}^{Q}, ..., \mathbf{h}_{w,N}^{Q}$, are projected to the LLM input dimension before sending to the LLM. Therefore, if the input sequence length of causal Q-Former is $T$, the number of sliding windows $W$ becomes $\lceil T/k \rceil$, and the overall output sequence length from causal Q-Former will be $W \times N$. Through end-to-end training, the output audio-visual representations of causal Q-Former are trained to align with the LLM input token space. Therefore, the use of sliding windows enables the LLM input token sequence length $W \times N$ to vary based on $T$ and can achieve a good trade-off between the degree of information reserved and the computation and storage costs.

Finally, the instruction prompt, such as questions or task descriptions will be appended to the concatenated output queries of all windows to form the input to the LLM. The response sequence $\hat{\mathbf{Y}}$ can be generated as follows:

$$\hat{\mathbf{Y}} = \underset{\mathbf{Y}}{\text{argmax}}\, P(\mathbf{Y}|\mathbf{h}_{1,1}^{Q}, \ldots, \mathbf{h}_{1,N}^{Q}, \ldots, \mathbf{h}_{W,1}^{Q}, \ldots, \mathbf{h}_{W,N}^{Q}, \mathbf{c}_1, \ldots, \mathbf{c}_M), \quad (3)$$

where $\mathbf{c}_1, \mathbf{c}_2, \ldots, \mathbf{c}_M$ are the contents of the prompt.

### 3.2 Q-FORMER WITH CAUSAL SELF-ATTENTION

The proposed causal Q-Former structure is shown in Fig. 2. To capture the causal temporal correlation among frames that are extracted independently, an additional causal self-attention module is added to the standard Q-Former structure, indicated by the red block in Fig. 2.

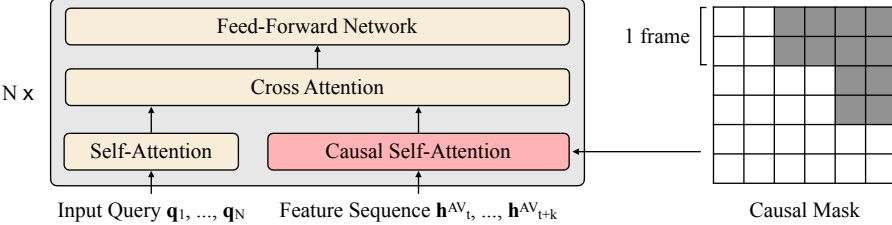

Figure 2: The causal attention module in the causal Q-Former with a block-wise triangular causal mask (grey cells are masked). The number of features per frame here is 2 as an example.

With the causal attention module, the encoding of one specific frame also includes the information of all previous frames carried in an auto-regressive way. This is particularly beneficial for causal reasoning questions, such as the "what happens next" questions (Xiao et al., 2021). Such questions are sometimes difficult to learn using only the positional embeddings.

### 3.3 System Training and Diversity Loss

The training data of video tasks, such as video question-answering (QA), usually only requires one or two keyframes, and the output queries tend to repeatedly capture the same information. Therefore, a novel diversity loss is proposed to encourage the causal Q-Former to extract more diverse aspects of the input sequence. Specifically, the diversity loss is formulated as:

$$\mathcal{L}_{\text{diverse}} = \sum_{w=1}^{W} \sum_{i=1}^{N} \sum_{j=1, j \neq i}^{N} \text{sim}(\mathbf{h}_{w,i}^{Q}, \mathbf{h}_{w,j}^{Q}), \quad (4)$$

where $W$ and $N$ are the total number of windows and the number of output queries of each window respectively, and $\text{sim}(\cdot)$ is the cosine similarity between two vectors. Cosine similarity is adopted since it is widely used for semantic similarity measurements, and in FAVOR, the output queries are aligned with a semantic space of the LLM input token representations. This choice is also supported

by the fact that the modulus of the output query tokens is very similar due to the layer normalisation operation of the causal Q-Former. By encouraging the audio-visual frames to be orthogonal to each other, the diversity loss forces the output query representations to be more spread in the text representation space. Overall, the system is trained in an end-to-end fashion using the cross-entropy (CE) loss and the diversity loss, as shown below:

$$\mathcal{L} = \mathcal{L}_{\text{CE}} + \lambda \mathcal{L}_{\text{diverse}}, \tag{5}$$

where $\lambda$ is the factor controlling the importance of the diversity loss, and the CE loss is calculated using the reference answer as the target.

## 4 EXPERIMENTAL SETUP

### 4.1 AUDIO-VISUAL EVALUATION BENCHMARK (AVEB)

In this paper, we ~~introduce~~propose the AVEB benchmark for audio-visual LLM evaluation, which evaluates single-modal perception ability via selected representative tasks while particularly focusing on multi-modal inference. AVEB contains 6 single-modal tasks, including automatic speech recognition (ASR) (Panayotov et al., 2015), audio captioning (AC) (Kim et al., 2019), image captioning (IC) (Young et al., 2014), optical character recognition (OCR) (Singh et al., 2019), visual question answer (VQA) (Hudson & Manning, 2019), and video question answer (Video QA) (Xu et al., 2017), together with 5 audio-visual tasks including audio-visual speech recognition (AVSR) (Sanabria et al., 2018), audio-visual scene-aware dialogue (AVSD) (Alamri et al., 2019), image spoken question answering (ISQA), audio-visual matching (AVM) (Hsu et al., 2020) and audio-visual sound source detection (AVSSD) (Chen et al., 2020; Zhao et al., 2023). ~~Related datasets are indicated in the citations. More details about the test datasets can be found in Appendix A.~~ In addition, we incorporate two widely used audio-visual benchmarks, the fine-grained audible video description (FAVD) (Shen et al., 2023) and the Vision-Audio-Language Omni-peRception (VALOR) (Chen et al., 2023c;d) benchmarks in our evaluation. Evaluation details can be found in Appendix B.

Table 1: AVEB details, including the number of samples used for evaluation and metrics reported. Since TextVQA, GQA, NExT-QA, AVSD and VGGSS test sets are large, randomly sampled subsets with enough samples for statistical significance were used in AVEB for efficient evaluation. The audio video matching part of AVM is zero-shot.

| Task | Test set | Num. of samples | Metrics | Zero-shot |
|------|----------|-----------------|---------|-----------|
| ASR | LibriSpeech test-clean | 2620 utterances | WER | No |
| AC | AudioCaps test | 938 audio clips | SPIDEr | No |
| IC | Flickr30k test | 1000 images | CIDEr / METEOR | Yes |
| OCR | TextVQA test | 1000 images | Accuracy | Yes |
| VQA | GQA testdev balanced | 1000 images | Accuracy | Yes |
| Video QA | NExT-QA test | 1000 clips | Accuracy | No |
| AVSR | How2 dev5 | 500 clips | WER | No |
| AVSD | AVSD val | 200 clips 2000 turns | Accuracy | No |
| ISQA | TextVQA + GQA | 2000 images | Accuracy | Yes |
| AVSSD | VGGSS | 850 video clips | Accuracy | Yes |
| AVM | SpokenCOCO val2014 + VGGSS | 1000 pairs 500 each | Accuracy | Yes |
| FAVD | FAVDBench | 1k videos | BLEU / METEOR | Yes |
| VALOR | VALOR 32k | 3k videos | CIDEr / METEOR | Yes |

~~ASR and AC are evaluated using word error rate (WER) and SPIDEr (Liu et al., 2017), a combination of SPICE and CIDEr respectively. The evaluation of IC uses CIDEr following (Dai et al., 2023), and METEOR, as LLMs tend to use a diverse range of words with similar meanings. OCR, VQA and Video QA are measured using top-1 accuracy. For OCR, the scoring follows (Singh et al., 2019) where each hit in the reference answer contributes 1/3 to the total hit. For VQA and Video QA, it is counted as correct if the reference answer exactly exists in the generated answer using a word-by-word matching. In particular, during inference only, Video QA is formulated as an in-context multiple-choice task where the choices are given in the prompt, and one hit is counted only when the generated answer exactly matches the reference. The same~~

measurement is taken for ISQA and AVM. Furthermore, for AVSD and AVSSD, as the reference answer is a full sentence, ChatGPT-assisted scoring is used to determine whether the generated answer is equivalent to the reference answer (see the prompt design in Appendix C).

While all other tasks already exist with open-source test sets, this paper particularly proposes ISQA and AVM tasks where audio-visual interaction is necessary. ISQA is the task where the question is in the audio and the answer can be found in the image. This test set is derived from the data used for OCR and VQA, where the questions are synthesised using a commercial text-to-speech synthesis system with a diverse range of speakers and styles. The text prompt is always "answer the question in the audio about the image", while the LLM is required to first understand the question in the speech, and then answer it by looking at the image. AVM is the task of determining whether the given spoken description in the SpokenCOCO dataset (Hsu et al., 2020) matches the image, or whether the given audio clip is compatible with the given video chosen from the VGGSS dataset (Chen et al., 2020). AVSSD is another task that requires a strong binding of audio and visual modalities, as a single modality usually only provides partial information about the sound.

## 4.2 MODEL CONFIGURATIONS

To validate the FAVOR learning framework, the Vicuna (Chiang et al., 2023) models (including 7B and 13B models, and 13B is the default option if not specified) are used as the LLM, Whisper (Radford et al., 2023) large-v2 encoder as the audio encoder and InstructBLIP (Dai et al., 2023) vision Transformer (ViT) plus Q-Former as the visual encoder. The visual encoder outputs 32 feature vectors for each video frame (every 0.5 seconds), and the audio encoder outputs 50 feature vectors per second. The causal Q-Former has two Transformer blocks with 768-dim hidden states. The output query representations are projected to 5120-dim before being sent to the LLM. The LLM is adapted using the low-rank adaptation (LoRA) (Hu et al., 2022) method with a rank of 32. Only the parameters of the attention query, key and value projections and feed-forward network weights are updated, which comprised 0.4% of the total number of LLM parameters.

Whisper and InstructBLIP are used as the single-modality baseline systems for comparison. As FAVOR adopted video data with different styles and focuses, to eliminate the discrepancy in training data and achieve fair comparisons, InstructBLIP is further fine-tuned on the same image and video training data as FAVOR. For each video clip, five equally-spaced frames were used resulting in 160 output queries. This is the same as the number of output queries used for 25-second videos in FAVOR. Video-LLaMA (Zhang et al., 2023b) was used as the multimodal baseline where only Vicuna-7B checkpoint was released for audio-visual input[1]. The VALOR-base model (Chen et al., 2023c) is used as the performance reference for the VALOR benchmark, as the total number of video samples to train FAVOR is only 1M. Note that VALOR-base is a BERT-based model and fine-tuned only on captioning tasks, which makes it not directly comparable to other multi-modal LLMs.

## 4.3 TRAINING DATA AND SPECIFICATIONS

FAVOR directly uses multi-task instruction fine-tuning to train the model parameters of causal Q-Former and LoRA. Training data contains both single-modal and audio-visual paired data. For audio-only tasks, LibriSpeech train-clean-100 and train-clean-360 sets are used for ASR, and AudioCaps are used for AC. For visual-only tasks. A mixture of LLAVA-150k (Liu et al., 2023) image QA data, OCRVQA OCR data (Mishra et al., 2019), TextCaps Sidorov et al. (2020) image caption data, NExT-QA video QA training data (Xiao et al., 2021), 5000 samples from COCO train2014 data with spoken captions (Lin et al., 2014) as well as 11k samples from VideoChat (Li et al., 2023b) are used. For audio-visual tasks, randomly selected 600-hour Ego4D video captioning data (Grauman et al., 2022), how2 300-hour training set AVSR data and AVSD training set are used. In order to further stimulate modality interactions during training, 5,000 images with spoken captions are used in the training set for the AVM task. Note that the entire training data only contains 1M samples with fewer than 300k video samples, and only contains publicly available datasets. Details about the training data can be found in Appendix A.

Furthermore, besides being trained using video and audio from the same source, FAVOR also uses randomly paired audio and video in training. This novel training approach increases versatility and achieves a better balance between the audio and visual modalities. It further enables FAVOR to perform audio-visual co-reasoning tasks as shown proposed in the AVEB benchmark, including ISQA and AVM. Moreover, we use a tiny storytelling set to further encourage a thorough mixture

---

[1] https://github.com/DAMO-NLP-SG/Video-LLaMA.git.

Table 2: AVEB single-modal task results. If specified, InstructBLIP is fine-tuned on the training data of FAVOR ("InstructBLIP fine-tuned"). IC is reported in CIDEr/METEOR. When using audio-only and visual-only inputs, the other modality is masked during training and inference. Tasks unable to be performed are marked with "-".

| Systems | ASR ↓ | AC ↑ | Video QA ↑ | IC ↑ | OCR ↑ | VQA ↑ |
|---|---|---|---|---|---|---|
| Whisper large-v2 | 2.9% | - | - | - | - | - |
| InstructBLIP 13B | - | - | 21.0% | 84.5 / 26.0 | 36.5% | **48.9**% |
| InstructBLIP 13B fine-tuned | - | - | 24.7% | 78.9 / 26.1 | 36.7% | 45.6% |
| Video-LLaMA 7B | - | - | 22.5% | 22.0 / 16.6 | 16.4% | 15.1% |
| FAVOR 13B (ours, audio-only) | **2.7**% | 39.7 | - | - | - | - |
| FAVOR 13B (ours, visual-only) | - | - | 44.8% | 74.0 / 26.5 | 34.2% | 45.6% |
| FAVOR 7B (ours, audio-visual) | 4.1% | 39.1 | 42.5% | 78.1 / 26.3 | 34.6% | 45.3% |
| FAVOR 13B (ours, audio-visual) | 3.3% | **42.6** | **49.3**% | **86.0 / 27.5** | **37.8**% | 45.2% |

Table 3: AVEB audio-visual task results. If specified, InstructBLIP is fine-tuned on the training data of FAVOR ("InstructBLIP†"). The other modality is masked in both training and testing when using audio-only and visual-only inputs. Tasks unable to be performed are marked with "-".

| Systems | AVSR ↓ | AVSD ↑ | ISQA ↑ | AVSSD ↑ | AVM ↑ |
|---|---|---|---|---|---|
| Whisper large-v2 | 8.3% | - | - | - | - |
| InstructBLIP 13B | - | 41.4% | - | 1.1% | - |
| InstructBLIP† 13B | - | 52.1% | - | 20.3% | - |
| Video-LLaMA 7B | - | 27.6% | - | 41.9% | 52.3% |
| FAVOR 13B (ours, audio-only) | 8.3% | - | - | 34.7% | - |
| FAVOR 13B (ours, visual-only) | - | 53.3% | - | 23.5% | - |
| FAVOR 7B (ours, audio-visual) | 8.7% | 51.2% | 24.5% | 50.5% | 74.3% |
| FAVOR 13B (ours, audio-visual) | **8.1**% | **54.5**% | **32.3**% | **51.1**% | **77.1**% |

of audio-visual descriptions for better demonstration quality only. ~~In addition to all the training datasets mentioned above, in order to explicitly encourage the model to generically combine both modalities, a storytelling fine-tuning set is designed. The dataset is gathered by prompting GPT-3.5 with reference audio caption or transcription, together with video caption, and asking GPT-3.5 to generate a coherent story combining both information (see details in Appendix D). The model is fine-tuned on this data for only 100 steps with a very small learning rate without causing any loss in the benchmark performance.~~

It is worth noting that in order to compare FAVOR with the original InstructBLIP on image tasks directly, Flickr30k for IC, TextVQA for OCR and GQA for VQA in the benchmark are not included in the training, and hence the model performed zero-shot learning on them. Moreover, since ISQA uses synthesised speech, this is also not a trained task and the model performed zero-shot learning.

## 5 EXPERIMENTAL RESULTS

### 5.1 MAIN RESULTS

The main results of using FAVOR on AVEB tasks are summarised in Table 2 and Table 3 for single-modal and audio-visual tasks respectively. While other models can only perform a subset of AVEB tasks, FAVOR is the first single model that achieves competitive performance on all tasks ~~compared to the single-modal counterparts~~, with remarkably better performance on audio-visual tasks. In particular, ~~as the first work that integrates audio, speech, image and video modality into LLMs,~~ FAVOR effectively achieves audio-visual co-reasoning which is reflected by the performance on ISQA, AVSSD and AVM tasks.

On audio-based tasks in Table 2, FAVOR obtains a similar WER compared to Whisper large-v2 and mixed results compared to the audio-only FAVOR. Further, with the aid of visual information, FAVOR achieves a lower WER on AVSR than both models in Table 3. On visual tasks, FAVOR demonstrates the best results on IC, OCR and Video QA, and on-par results on VQA with Instruct-BLIP fine-tuned on the same training set. In particular, the fine-grained causal modelling of video in FAVOR yields over 20% improvements compared to InstructBLIP even though the latter is fine-tuned on the same set of video data.

Table 4: Results on FAVDBench (BLEU1/BLEU4/METEOR) and VALOR (METEOR/CIDEr) tasks. **Our fine-tuning on VALOR is performed only on 10% of the VALOR training data.**

| Systems | FAVD ↑ | FAVD fine-tuned ↑ | VALOR ↑ | VALOR fine-tuned ↑ |
|---|---|---|---|---|
| VALOR-base† | - | - | - | **14.8 / 55.7** |
| Video-LLaMA 7B | 20.8 / 2.4 / 15.0 | 39.4 / 6.5 / 16.5 | **10.7** / 1.2 | 10.9 / 21.3 |
| FAVOR 7B (ours) | 24.9 / 2.8 / 14.8 | 42.6 / 9.9 / 18.3 | 8.6 / 13.4 | 13.9 / 42.6 |
| FAVOR 13B (ours) | **28.2 / 3.0 / 15.2** | **44.2 / 10.9 / 19.1** | 8.8 / **15.3** | 14.2 / 46.9 |

Table 5: Ablation studies on the core components of FAVOR based on video and audio-visual tasks. Each row represents removing one component with other parts remaining the same. Note the last row is equivalent to Video-LLaMA with high temporal resolution, speech encoder and LoRA, and the comparison to complete FAVOR directly reflected the benefit of the proposed structure design.

| Systems | Video QA | AVSR | AVSD | ISQA | AVSSD | AVM |
|---|---|---|---|---|---|---|
| Complete FAVOR | **49.3**% | 8.1% | **54.5**% | **32.3**% | **51.1**% | **77.1**% |
| FAVOR without causal encoder | 42.8% | **8.0**% | 54.1% | 20.9% | 37.1% | 74.8% |
| FAVOR without sliding window | 44.8% | 8.5% | 53.6% | 29.7% | 45.3% | 74.5% |
| FAVOR without synchronisation | 47.4% | 8.4% | 53.4% | 17.2% | 50.5% | 72.5% |
| FAVOR without causal encoder, diversity loss, and synchronisation | 41.8% | 8.9% | 50.5% | 16.7% | 38.6% | 72.0% |

On the audio-visual tasks in Table 3, while outperforming all the baseline systems in every task, FAVOR demonstrated a strong audio-visual co-reasoning ability based on the audio-visual matching (AVM) dataset results and is the only system to our knowledge that can perform speech-image co-reasoning based on image-spoken QA (ISQA). Audio-visual co-reasoning (including speech-image co-reasoning) is an important yet challenging ability which requires the model to comprehend the visual content as well as both speech and non-speech sounds in the audio, and to capture the correlation between what it "hears" and "sees". Such tasks were almost infeasible for any other audio-visual models so far, since they were unable to understand both speech and non-speech sounds and did not model the audio-visual correlations in fine-grain. Various audio-visual emergent abilities in addition to the audio-visual co-reasoning ability, as discussed in Section 5.5.

Results on FAVD and VALOR test data (Table 4) also demonstrated the superiority of FAVOR over Video-LLaMA. In the zero-shot case, Video-LLaMA tends to generate long paragraphs of text even under the instruction of generating short sentence responses. This resulted in extremely low CIDEr scores compared to FAVOR which closely follows the instruction and generate concise responses. Notably, the best FAVOR model achieves better performance on FAVD than the best value reported in (Shen et al., 2023). Although FAVOR uses only 10% of the VALOR training data for fine-tuning, it achieves competitive performance on the VALOR test data.

## 5.2 ABLATION STUDIES

Detailed ablation studies are performed for each proposed component in FAVOR as shown in Table 9 for single-modal tasks and Table 10 for multimodal tasks in Appendix E. This section particularly focuses on the use of causal Q-Former and audio-visual synchronisation on video and audio-visual tasks, as summarised in Table 5.

First, the effect of the causal attention module is most clearly reflected by the performance on video QA, ISQA and AVSSD, as it both boosted the temporal causality modelling as well as provided a better audio-visual fusion before applying the cross-attention in the Q-Former. Second, the fine-grained model design, including sliding windows and frame-level synchronisation ~~the use of a sliding window~~ is crucial to achieving good results on the AVSR task ~~speech input as shown in the AVSR results~~. Without the sliding window, a fixed number of output queries are used no matter how long the audio is, which results in more deletion errors. Besides, using sliding windows also benefits the video QA task as they encourage the localised causal relationship to be captured. Furthermore, the use of synchronisation is crucial for audio-visual co-reasoning to work as supported by the ISQA and AVM results. Without synchronisation, modality alignment is done rather independently and the correlation between audio and video is only modelled among high-level features

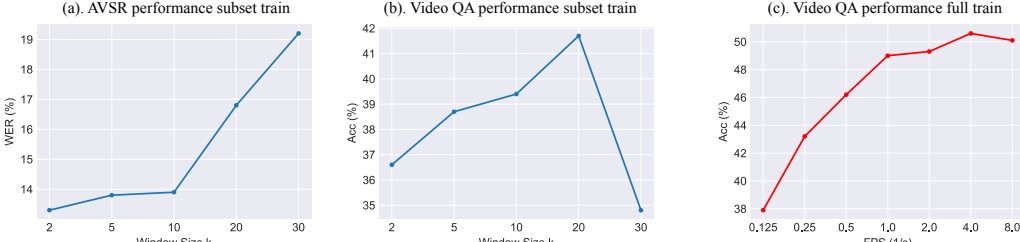

Figure 3: Influence of the window sizes and the frames per second (FPS) to the model performance on speech and video tasks. (a) and (b): results by training and evaluating using different window sizes $k$ on 10% of data. (c): the influence of FPS using the best model on full data.

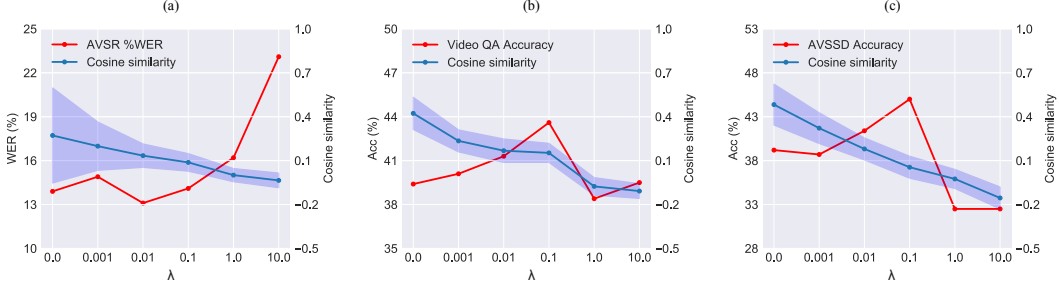

Figure 4: Variations of model performance due to the diversity loss factor, *i.e.* $\lambda$ in Eqn. (4), on (a) AVSR measured in %WER, (b) Video QA measured in %Accuracy and (c) AVSSD measured in %Accuracy. Variations of average cosine similarities are also shown under different $\lambda$'s.

that are aligned in the text space. This may easily omit information about the concurrency of audio and visual contents, such as how a specific part of speech relates to a specific visual scene. On the other hand, synchronisation enables a temporally aligned cross-modal interaction which allows such concurrency to be captured, resulting in enhanced performances on audio-visual tasks.

## 5.3 Analysis on the Sliding Windows and Temporal Resolution

As mentioned in Section 3.1, the trade-off between the sliding window size and the model performance is shown in Figure 3. Specifically, (a) and (b) show the influence of the numbers of frames $k$ in a window while keeping the ratio $N/k$ a constant (*i.e.* keeping the total output queries $W \times N$ unchanged) and the same frame rate. This is trained on 10% of the full training data for quick experiments. Although using shorter windows benefits ASR, as fewer output tokens are used to encapsulate all the visual information within that window, performance on video QA is degraded. On the other hand, larger windows heavily reduce the ASR performance as the monotonic alignment in ASR is especially difficult to learn with 10% of the training data.

Figure 3 (c) clearly shows the importance of high temporal resolution in video modelling. The lowest FPS is equivalent to 8 frames per video, *e.g.* Video-LLaMA, where over 24% relative accuracy improvements are achieved using an FPS of 2. ~~Figure 3 (c) shows the influence of the number of frames per second (FPS) on the model performance during inference.~~ The best model trained on the full set is used with the same number of frames per window. While low accuracy is observed when the frame rate is low, increasing FPS beyond 1.0 only receives marginal improvements at the cost of having many more output queries sent to the LLM. 2.0 FPS was chosen as it made the audio and visual sequences have the most similar lengths, and hence easier for synchronisation.

## 5.4 Analysis of the Diversity Loss

Analysis of the effect of diversity loss is also performed using 10% of the training data as shown in Figure 4, and examples of cosine similarity matrices among output queries are shown in Appendix F. For ASR, the model is trained to include all the speech information in the audio sequence and the cosine similarity varies according to the length of the speech. For videos, the cosine similarity is close and does not vary too much for different video lengths, and hence diversity loss effectively acts as a way to encourage more diversified information to be captured. However, when a high

$\lambda$ is employed, diverse information causes confusion in the model and results in a more severe hallucination problem (*e.g.* high insertion rate in WER) with heavily degraded model performance.

### 5.5 DISCUSSIONS ON INCORPORATING SPEECH AND SPEECH-VIDEO INTERACTIONS

Speech is an important source of information for video that should always be considered for audio-visual LLM to perform a comprehensive understanding. Unlike audio events, the speech content can hardly be inferred from the visual modality, making it particularly indispensable to comprehend any videos involving people talking. Moreover, the co-occurrence of speech and video events, which is modelled by the fine-grained temporal synchronisation in FAVOR, is required to understand the audio-visual temporal relations, *e.g.* "What did A say" (more examples in Appendix G).

One of the major contributions of FAVOR is to incorporate speech in a multimodal LLM and effectively combine both speech and video content to generate responses. In addition to the ISQA and AVM tasks that have already reflected the co-reasoning ability, the advantage of FAVOR can be more clearly demonstrated by the emergent abilities (shown in Appendix G). For instance, in response to questions about why a movie clip is funny or romantic, FAVOR combines the video, dialogue between characters and background audio or music to generate a more encompassing and convincing answer. Besides, FAVOR is able to understand the scene better by using knowledge from the speech, such as the species of a particular fish introduced in a documentary.

## 6 CONCLUSION

This paper proposed FAVOR, a fine-grained audio-visual joint representation learning framework for multimodal LLMs. On the introduced~~proposed~~ AVEB benchmark for audio-visual evaluation, FAVOR achieved competitive performance on audio and visual single-modal tasks with a remarkable 20% absolute accuracy improvement on the causal reasoning video QA task compared to the baselines. FAVOR demonstrated audio-visual, and particularly strong speech-visual co-reasoning abilities, with remarkable cross-modal emergent abilities demonstrated via examples.

## 7 REPRODUCIBILITY STATEMENT

To make the experiments and models reproducible, the benchmark details are provided in the supplementary materials, and a demo page is provided in the abstract for a convenient try-out of the model. The details of the training and test data are summarised in Section 4 and Appendix A. Key hyper-parameter settings were discussed in the result section. The complete training and inference code together with model checkpoints will be released upon acceptance.

## 8 ETHICAL STATEMENT

The approaches in this paper do not give rise to any additional risks beyond the ones directly inherited from the model checkpoints. The ASR encoder and visual encoder might work worse for people from particular demographics. The framework also inherits the biases from all the large language models used for experiments in this paper.

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

# A    TRAINING SET AND BENCHMARK DETAILS

| Data | In Train | In AVEB | Description |
|------|----------|---------|-------------|
| LibriSpeech | Yes | Yes | LibriSpeech is an English audiobook data. The train-clean-100 and train-clean-360 splits were used for training, and test-clean was used in AVEB. Prompt example: "Transcribe the speech into text." |
| AudioCaps | Yes | Yes | AudioCaps is a widely used audio caption dataset containing 46k 10-second audio samples with manually annotated captions. Example prompt: "Please describe the audio." |
| LLAVA-150k | Yes | No | LLAVA-150k contain QA pairs generated using ChatGPT. Example prompt: "What does the man hold in the image?" |
| OCRVQA | Yes | No | OCRVQA is an OCR-based QA dataset containing questions mostly about printed words in an image. Example prompt: "Who wrote this book?" |
| TextVQA | No | Yes | OCR-based QA dataset containing questions about various words in realistic scenes (*c.f.* printed words). Example prompt: "What is the brand of this camera?" |
| Flickr30k | No | Yes | Image caption dataset where each image is annotated with manual single-sentence descriptions. Example prompt: "Describe this image in one short sentence." |
| GQA | No | Yes | GQA consists of questions about various day-to-day real-world images. This involves reasoning skills about the objects in the image. Example prompt: "What kind of device is on top of the desk?" |
| TextCaps | Yes | No | Image caption data particularly focusing on capturing text in the image. Only 80k samples were randomly selected for training. Example prompt: "Describe the image." |
| MSVD-QA | Yes | No | MSVD-QA is a dataset with questions about real-world video clips. Example prompt: "In the video, what is the man with long hair playing?" |
| NExT-QA | Yes | Yes | NExT-QA is a video QA dataset, particularly focusing on causal and temporal correlations. Example prompt: "What does the girl in white do after bending down in the middle? Options/Choose one from: (Add choices here during inference)". |
| VideoChat | Yes | No | A GPT4-generated video QA dataset where the question mainly asks for detailed descriptions of the video. Example prompt: "Provide a detailed description of the given video." |
| AVSD | Yes | Yes | Audio-visual scene-aware dialogue data where questions are raised in turns about the video and the audio in the video. Example prompt: "And then what happened?" and "Is the man saying anything?" |
| Ego4D | Yes | No | An audio-visual dataset containing egocentric videos. Video descriptions were used as supervision signals which came from single-sentence short clip descriptions that were concatenated and refined using ChatGPT. Example prompt: "Describe the video in detail." |
| How2 | Yes | Yes | An audio-visual speech recognition dataset containing videos explaining how to perform various tasks. Example prompt: "Transcribe the speech into text, paying attention to both audio and video." |
| VGGSS | No | Yes | Sound source localisation data containing questions about the sound source in a 5-to-10-second video clip. Example prompt: "What is the source of the sound?" |

Table 6: Dataset and benchmark details

A range of datasets spanning audio and visual tasks were used in our experiments. Table 6 summarises these datasets in detail, with individual descriptions and relevant prompt designs.

## B  EVALUATION DETAILS

ASR and AC are evaluated using word error rate (WER) and SPIDEr (Liu et al., 2017), a combination of SPICE and CIDEr respectively. The evaluation of IC uses CIDEr following (Dai et al., 2023), and METEOR, as LLMs tend to use a diverse range of words with similar meanings. OCR, VQA and Video QA are measured using top-1 accuracy. For OCR, the scoring follows (Singh et al., 2019) where each hit in the reference answer contributes 1/3 to the total hit. For VQA and Video QA, it is counted as correct if the reference answer exactly exists in the generated answer using word-by-word matching. It is needed to check the opposite answer doesn't exist for yes-or-no questions. In particular, during inference only, Video QA is formulated as an in-context multiple-choice task where the choices are given in the prompt, and one hit is counted only when the generated answer exactly matches the reference. The same measurement is taken for ISQA and AVM. Furthermore, for AVSD and AVSSD, as the reference answer is a full sentence, ChatGPT-assisted scoring is used to determine whether the generated answer is equivalent to the reference answer (see the prompt design in C).

## C  GPT SCORING PROMPT DESIGN

As open-ended questions in AVSD and VGGSS datasets contain full-sentence answers rather than one or two words, it is difficult to evaluate via string matching. Therefore, ChatGPT was used to assist with the evaluation. Prompt designs for each task are described in Table.

| Task | Description |
|------|-------------|
| AVSD | Given the question "QUESTION", is the answer "HYPOTHESIS" equivalent to the answer "REFERENCE"? Answer "Yes" if they are equivalent, and "No" if they are different. |
| VGGSS | Is the sound source mentioned in answer "REFERENCE" the same as the sound source mentioned in answer "HYPOTHESIS"? Answer "Yes" if they are the same, and "No" if they are different or one does not mention the sound source. |

Table 7: Prompt design for ChatGPT-based evaluation. Note that QUESTION refers to the question, HYPOTHESIS is the model-generated answer and REFERENCE is the reference answer.

## D  STORYTELLING DATASET

To better stimulate the ability to generate long descriptions using both audio and visual input, a tiny storytelling dataset was generated to further fine-tune the model for a small number of steps. This was generated by prompting GPT-3.5 with audio descriptions (including reference captions and reference speech transcriptions) as well as reference video descriptions. Audio samples were randomly selected from both AudioCaps and LibriSpeech datasets, while video samples were randomly selected from the Ego4D dataset. The prompt and the generated samples are summarised in Table 8.

| Prompt | Audio caption or speech transcription: AUDIO CAPTION. Video description: VIDEO DESCRIPTION. Write a story combining both audio and video information. |
|--------|-------------|
| Data sample | USER: <FAVOR output query representations> Write a story about what you hear and see. LLM: Once apon a time, ... |

Table 8: Prompt to generate the storytelling fine-tuning set and example training data.

# E    ABLATION STUDIES

The full ablation studies on all tasks in AVEB are shown in Table 9 and 10, as complementary materials for section 5.2

Table 9: Ablation studies on the AVEB single-modal tasks.

| Systems | ASR ↓ | AC ↑ | IC ↑ | OCR ↑ | VQA ↑ | Video QA ↑ |
|---|---|---|---|---|---|---|
| FAVOR | 3.3% | 42.6 | 86.0 / 27.5 | 37.8% | 45.2% | 49.3% |
| without diversity loss | 3.1% | 42.7 | 71.9 / 26.1 | 37.2% | 46.2% | 47.1% |
| without causal enc. | 3.0% | 44.0 | 72.1 / 26.0 | 34.9% | 45.7% | 42.8% |
| without sliding window | 3.3% | 42.7 | 76.8 / 26.4 | 34.6% | 44.8% | 44.8% |
| without synchronisation | 3.0% | 40.6 | 85.6 / 26.7 | 32.5% | 46.1% | 47.4% |
| without causal encoder, diversity loss, and synchronisation | 3.1% | 36.0 | 71.9 / 26.0 | 34.7% | 44.8% | 41.8% |

Table 10: Ablation studies on the AVEB audio-visual tasks.

| Systems | AVSR ↓ | AVSD ↑ | ISQA ↑ | AVSSD ↑ | AVM ↑ |
|---|---|---|---|---|---|
| FAVOR | 8.1% | 54.5% | 32.3% | 51.1% | 77.1% |
| without diversity loss | 8.1% | 53.9% | 34.1% | 53.5% | 75.7% |
| without causal enc. | 8.0% | 54.1% | 20.9% | 37.1% | 74.8% |
| without sliding window | 8.5% | 53.6% | 29.7% | 45.3% | 74.5% |
| without synchronisation | 8.4% | 53.4% | 17.2% | 50.5% | 72.5% |
| without causal encoder, diversity loss, and synchronisation | 8.9% | 50.5% | 16.7% | 38.6% | 72.0% |

# F    VISUALISATION OF DIVERSITY LOSS EFFECT

The cosine similarities among output query representations of the causal Q-Former under different diversity loss factors are shown in Fig. 5.

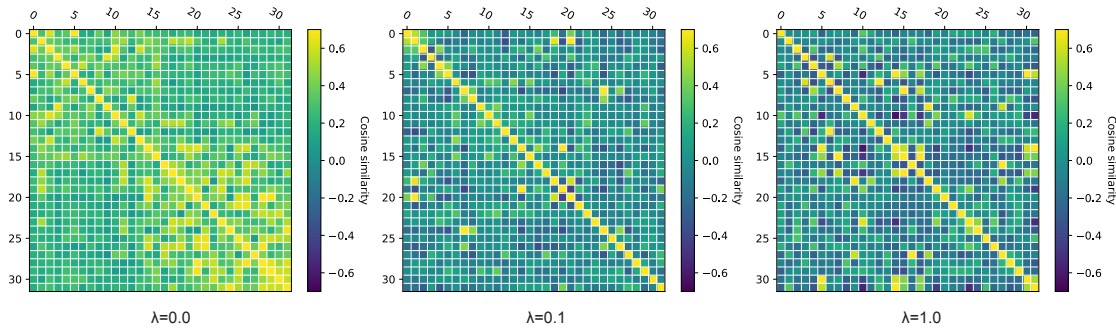

Figure 5: Visualisation of cosine similarity matrix with different diversity loss factors.

# G    CASE STUDIES

Six cases are illustrated in Fig. 6 to Fig. 12.

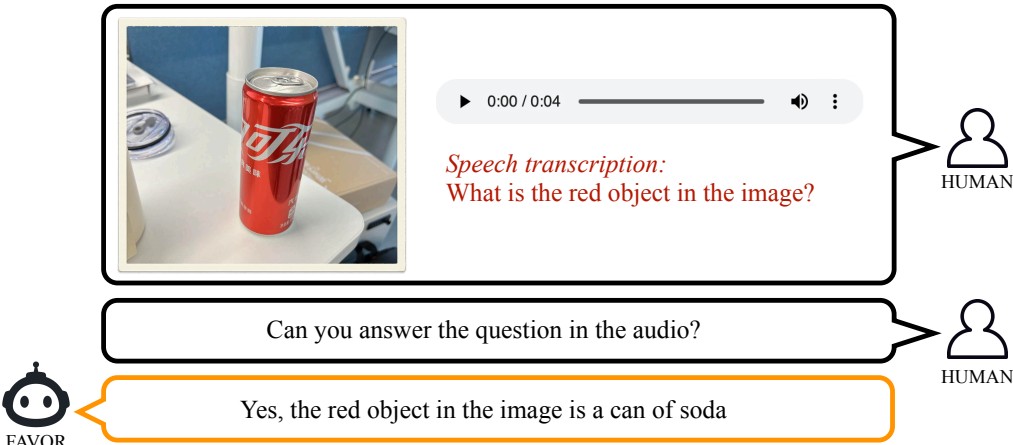

Figure 6: Case study 1 – Image-spoken QA task with real audio.

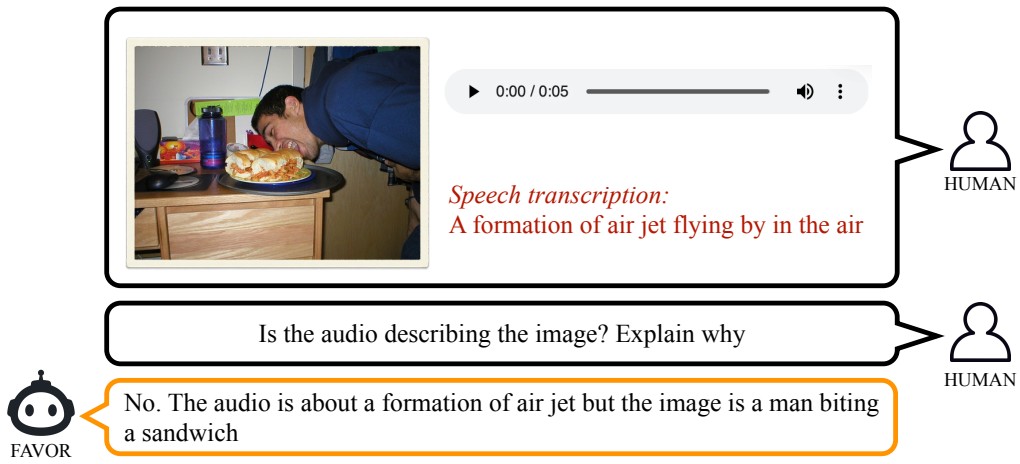

Figure 7: Case study 2 – Audio-visual matching task with the request for explanation. During the benchmark test, the explanation was removed. The answer shows the understanding of both the speech and the image as well as the ability to perform reasoning based on them.

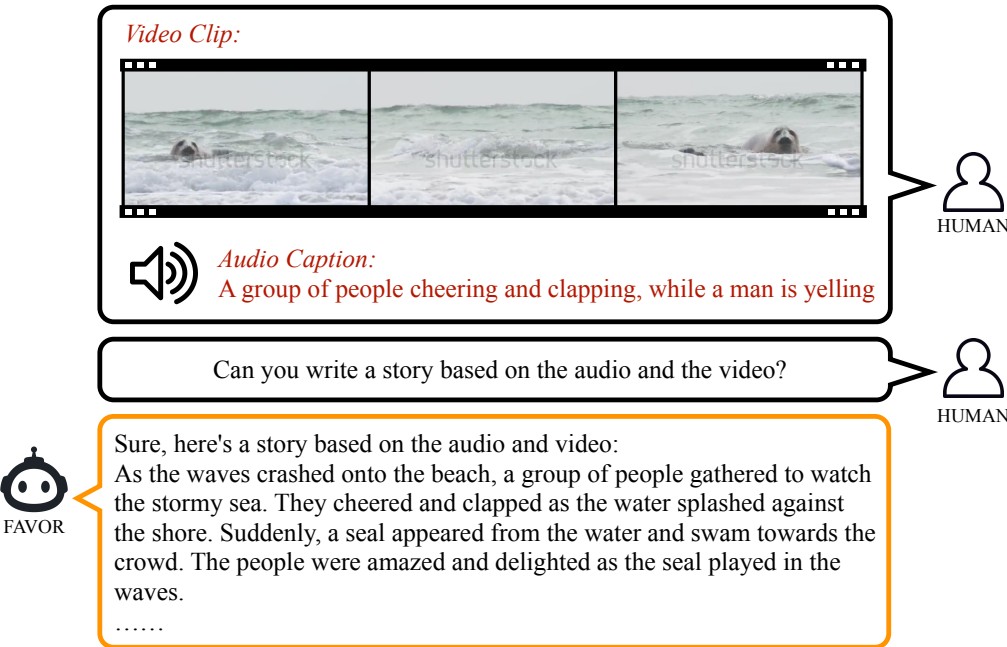

Figure 8: Case study 3 – Storytelling task with a video clip and the audio came from a different source. The answer combines the audio event, such as cheering and clapping, coherently with the video content, such as the seal.

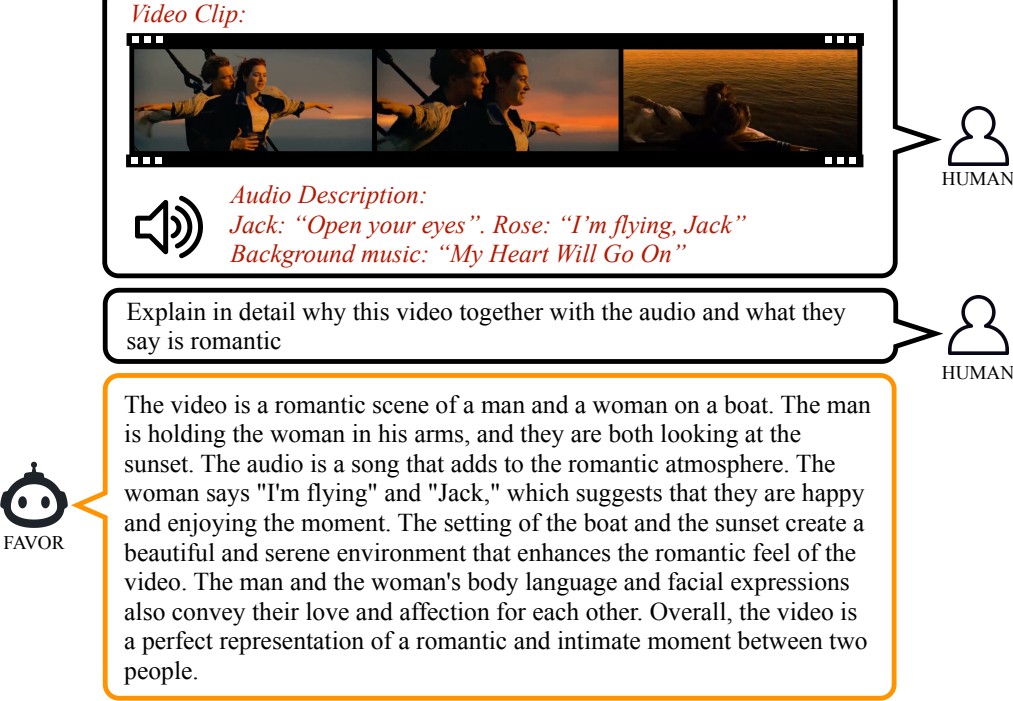

Figure 9: Case study 4 – The famous scene in the movie *Titanic* could be understood by FAVOR. The understanding combines the visual scene, the dialogue between characters, *e.g.* "I'm flying, Jack", as well as the background music to make the response comprehensive. It also reflected that the system knows the speaker by quoting the heroine's speech.

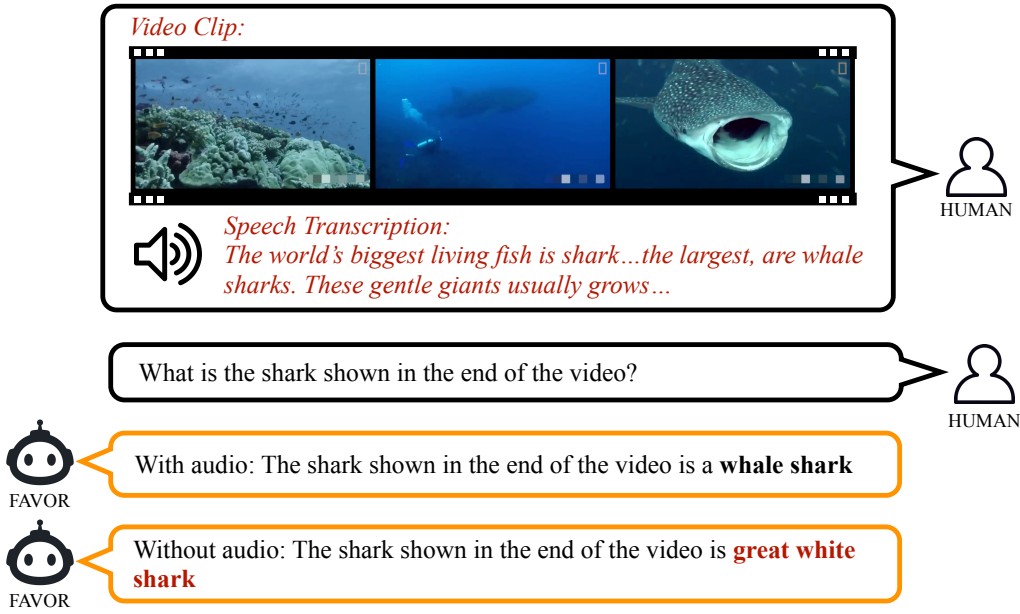

Figure 10: Case study 5 – Demonstration of how speech content could provide knowledge for visual understanding. The system was clearly unable to identify the species of the shark without the help of the audio, and just made the most likely guess.

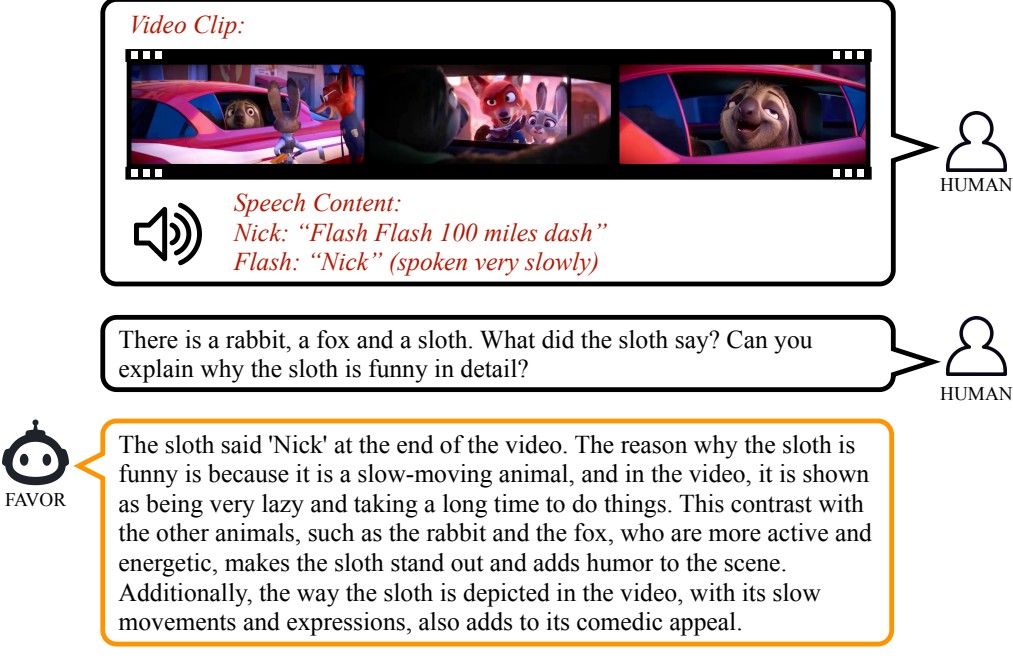

Figure 11: Case study 6 – Demonstration of understanding cartoon clips about the amusing sloth character named "Flash" in *Zootopia*. FAVOR explained using both audio and video, and accurately attributed the word "Nick" to the sloth.

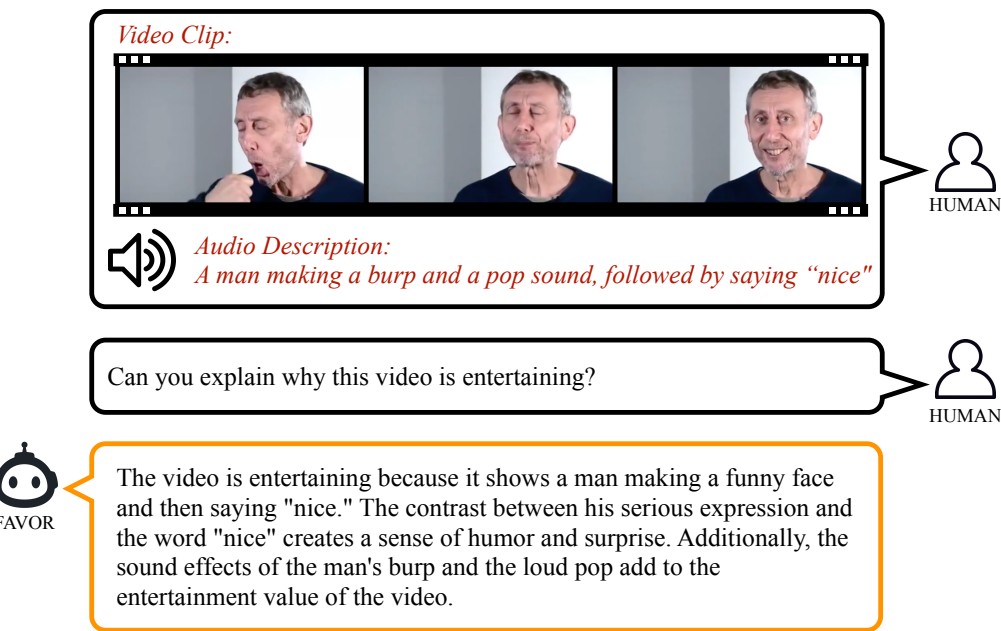

Figure 12: Case study 7 – Demonstration of FAVOR using audio, speech and video to explain why a specific meme is interesting. The explanation includes the funny sound, the word being said with the facial expression.

