# OpenReview forum: "FINE-GRAINED AUDIO-VISUAL JOINT REPRESENTATIONS FOR MULTIMODAL LARGE LANGUAGE MODELS"
_ICLR.cc/2024/Conference — Submitted to ICLR 2024_

### Official Review · Reviewer_QdMV · 2023-10-21

**Soundness:** 3 good
**Presentation:** 4 excellent
**Contribution:** 2 fair
**Rating:** 6
**Confidence:** 3

**Summary:**

The goal of this paper is to introduce an architecture which leverages large language models to solve multi-modal tasks. The authors do this by using 3 modalities: visual, audio and text. They introduce a a Temporal synchronization module that synchronize the visual and audio modalities. Moreover, they introduce a variant of Q-Former which uses causal attention.

**Strengths:**

- The paper is well organised and it is very easy to be read and understood.
- The method obtains significant improvements over the previous systems.

**Weaknesses:**

- Very personal opinion for Fig 1: I think the trainable modules should have different colors to be even easier to understand the figure. All causal Q-former should have the same color, as they are applied on multiple windows with the same parameters. But then, the LLM should have a different color and each encoder should also have a different color. In this way you are making sure no one thinks you are using the same parameters across all these. Also a lot of papers use a special symbol (the fire symbol) for the trainable layers, as in video-llama figure and other papers which I think would help if you also used it.
- The novelty is somewhat limited as I will argue below:
    1. Regarding the benchmarks, what is the novelty? As I understand, the paper does not provide a newly created dataset, but only puts together previously established datasets and calls it a new benchmark. Correct me if I am wrong.
     2.  As I could see, the only difference between the proposed method and the video-LLAMA is using causal attention in the q-former, and the synchronization.
- Why some datasets were chosen only for training and some only for testing? In the paper, there are some explanations for some of them, but there are still datasets such as llava-150k, text caps, ego4d or videochat, that were chosen for either testing or training without mentioning why.
- From section 4.2 :“fair comparisons, InstructBLIP is further fine-tuned on the same image and video training data as FAVOR.” Was the same done for Whisper? If not, why? If yes, mention it.
- Does Video-LLAMA use the same exact visual/audio encoders? If not, I am afraid the results will not reflect the true capabilities of this model.
- Moreover, Video-LLAMA does not finetune the LLM, while this paper uses LoRA to finetune it. While one can argue that LoRA does not actually change the original weights, there is still some training happening there. Thus, I would also ask the authors to do an experiment where they keep the LLM frozen, as in video-llama and just train the other components. In this way it will be clear if the benefits come from the causal attention/synchronization model or if they come just from the fact that there are many more trainable parameters in the proposed model.
- In Table 2 and 3, some results are missing for the Video-LLAMA and are replaced with “-”. Why? As far as I understand, Video-LLAMA could be applied on all the tasks that the proposed method can be applied to by just removing the missing modality. One can just remove the audio or the visual branch in the Video-LLAMA and turn it into a uni-modal model. One would not even need to re-train it as the LLM is not trained at all, so it does not “care” if there are 2 modalities as input or only one. Please, correct me if I am wrong.

**Questions:**

- My main issue is the lack of novelty and the issue that the results may be good due to “unfair decisions”.
- Mainly, the authors need to clarify if both the proposed model and the video-llama use the same encoders or not. If they don’t use the same encoders, the experiments should be rerun to have a consistent setting with the same encoders.
- Moreover, the authors need to provide evidence that even when not fine-tuning the LLM and using consistent encoders with video-llama, their causal Q-former and synchronization module helps. Right now, the boost in performance could be attributed entirely to fine tuning the LLM.
- The authors should run the Video-LLAMA also on the task marked “-”, as mentioned in my previou comments. However, as I may be wrong here, I ask for clarifications for why this can not be done, if it can really not be done.
- If these issues are addressed (mainly the fairness comparison), I think this paper is a good one, and even if it does not have a lot of novelty, it proposes a fairly simple extension that can obtain a significant boost in performance. But as it stands right now, there seems to be an unfair comparison and also lack of novelty and the paper is not good enough for being accepted.

---

> ### Author Response · Authors · 2023-11-15
> **Response to Reviewer QdMV**
>
> We’d like to thank the reviewer for the positive comments and constructive suggestions and would like to respond to each individual concern.
>
> - Weakness 1:
> > Very personal opinion for Fig 1: I think the trainable modules should have different colors to be even easier to understand the figure. All causal Q-former should have the same color, as they are applied on multiple windows with the same parameters. But then, the LLM should have a different color and each encoder should also have a different color. In this way you are making sure no one thinks you are using the same parameters across all these. Also a lot of papers use a special symbol (the fire symbol) for the trainable layers, as in video-llama figure and other papers which I think would help if you also used it.
>   - Thanks for pointing this out. We have added the fire symbol and made different blocks have different colours in Fig. 1.
>
> - For weakness 2, we would like to first clearly list FAVOR’s contribution as follows:
>   - FAVOR is the first single model that integrates audio, speech and video inputs, which is the most obvious difference between FAVOR and other multimodal LLMs. The incorporation of speech causes significant changes both to the approach itself and the extent of applications.
>
>   - Unlike audio events, speech contains more temporal fine-grained information that is closely correlated with but very difficult to directly infer from video. This nature motivates the design of the FAVOR structure, including fine-grained (high temporal resolution) modelling, audio-visual synchronisation, causal attention module, and diversity loss.
>
>   - Instead of having audio and video streams coming from the same source, FAVOR in addition supports audio from a different source which enables the training of the two modalities more balanced and also the performing of more versatile audio-visual co-reasoning tasks, including ISQA and AVM.
>
>   - To the best of our knowledge, FAVOR is the first model that is able to perform open-ended audio-visual-speech-text questions, such as those given in Fig. 9 to 12 in the appendix.
>
> - For each point of weakness 2:
> > Regarding the benchmarks, what is the novelty? As I understand, the paper does not provide a newly created dataset, but only puts together previously established datasets and calls it a new benchmark. Correct me if I am wrong.
>   - We would like to clarify that the AVEB benchmark also contains newly designed tasks to reflect the audio-visual co-reasoning ability, including ISQA and AVM. These require the system to understand speech.
>
>   > As I could see, the only difference between the proposed method and the video-LLAMA is using causal attention in the q-former, and the synchronization.
>   - While FAVOR has some similarities to Video-LLaMA, we’d like to clarify the following superior aspects of FAVOR:
>
>     - FAVOR is able to understand speech input which is unable to be handled by Video-LLaMA.
>
>     - Compared to Video-LLaMA, incorporating speech input motivated us to design this significantly different model structure of FAVOR, including the synchronisation, sliding window operation and causal Q-Former.
>
>     - In contrast to sampling a fixed number of frames regardless of the video length in Video-LLaMA, FAVOR performs fine-grained modelling and the sliding window allows more output tokens for longer videos, which leads to significantly better performance.
>
>   - __As a result, Video-LLaMA could not perform any of the examples shown in Fig. 6 to 12.__
>
> - Weakness 3
> > Why some datasets were chosen only for training and some only for testing? In the paper, there are some explanations for some of them, but there are still datasets such as llava-150k, text caps, ego4d or videochat, that were chosen for either testing or training without mentioning why.
>   - Tasks including IC, OCR, VQA and AVSSD are set as zero-shot tasks for a fair comparison with InstructBLIP and Video-LLaMA. ISQA and AVM are set to be zero-shot tasks due to the limited data samples.
>
> - Weakness 4:
> > From section 4.2 :“fair comparisons, InstructBLIP is further fine-tuned on the same image and video training data as FAVOR.” Was the same done for Whisper? If not, why? If yes, mention it.
>   - We did not perform fine-tuning on Whisper as the ASR training data used here is small, and fine-tuning usually leads to degraded performance. The main aim of fine-tuning InstructBLIP is to eliminate the training data variable and fairly compare the structure advantage of FAVOR.

---

> > ### Author Response · Authors · 2023-11-15
> > **Response to Reviewer QdMV Part 2**
> >
> > Continued from the response above:
> > - For weaknesses 5 and 6:
> > > Does Video-LLAMA use the same exact visual/audio encoders? If not, I am afraid the results will not reflect the true capabilities of this model.Moreover, Video-LLAMA does not finetune the LLM, while this paper uses LoRA to finetune it. While one can argue that LoRA does not actually change the original weights, there is still some training happening there. Thus, I would also ask the authors to do an experiment where they keep the LLM frozen, as in video-llama and just train the other components. In this way it will be clear if the benefits come from the causal attention/synchronization model or if they come just from the fact that there are many more trainable parameters in the proposed model.
> >   - Regarding a fair comparison with Video-LLaMA, because incorporating speech is the key innovation, choosing a speech-based encoder is one of the important aspects of the model design. Therefore, it is reasonable to compare it to the Video-LLaMA in its original form although having a different audio encoder. However, we could still provide a point of comparison in response to your suggestions of modifying Video-LLaMA, by replacing the encoder, changing the video frame rate and using LoRA for training. This is the last row of Tables 8 and 9. The comparison to the last row clearly shows __over 10% improvements__ across all audio-visual tasks using synchronisation and causal Q-Former.
> >
> > - Weakness 7
> > > In Table 2 and 3, some results are missing for the Video-LLAMA and are replaced with “-”. Why? As far as I understand, Video-LLAMA could be applied on all the tasks that the proposed method can be applied to by just removing the missing modality. One can just remove the audio or the visual branch in the Video-LLAMA and turn it into a uni-modal model. One would not even need to re-train it as the LLM is not trained at all, so it does not “care” if there are 2 modalities as input or only one. Please, correct me if I am wrong.
> >   - Video-LLaMA can not perform ASR and AVSR (WER over 100%), and hence also ISQA. Video-LLaMA is not trained to support single audio input, and hence single audio input will confuse the model.
> >
> > I believe the questions are a summary of the weaknesses. Without repeating, we’d like to provide the following answers to each question:
> > - Questions 1 and 2:
> > > My main issue is the lack of novelty and the issue that the results may be good due to “unfair decisions”. Mainly, the authors need to clarify if both the proposed model and the video-llama use the same encoders or not. If they don’t use the same encoders, the experiments should be rerun to have a consistent setting with the same encoders.
> >   - As explained for weaknesses 3 to 6, we believe our experiments are fair and the required comparison point (including different encoders and LoRA) can be found in Tables 8 and 9.
> >
> > - Question 3:
> > > Moreover, the authors need to provide evidence that even when not fine-tuning the LLM and using consistent encoders with video-llama, their causal Q-former and synchronization module helps. Right now, the boost in performance could be attributed entirely to fine tuning the LLM.
> >   - As explained for weaknesses 5 and 6, we’d like to further emphasise that 1). Incorporating speech and using the speech encoder is one of the key contributions. 2). The improvements using synchronisation and causal Q-Former is shown in Table 8 and 9. 3). It is also worth pointing out that Video-LLaMA used the entire 2.5M WebVid videos to train the video Q-Former which is 10 times more video data than what we used for training.
> >
> > - Question 4:
> > > The authors should run the Video-LLAMA also on the task marked “-”, as mentioned in my previou comments. However, as I may be wrong here, I ask for clarifications for why this can not be done, if it can really not be done.
> >   - Please find our explanation for weakness 7 on this problem.
> >
> > - Question 5:
> > > If these issues are addressed (mainly the fairness comparison), I think this paper is a good one, and even if it does not have a lot of novelty, it proposes a fairly simple extension that can obtain a significant boost in performance. But as it stands right now, there seems to be an unfair comparison and also lack of novelty and the paper is not good enough for being accepted.
> >   - Thank you for the positive comments, and hope our explanation resolves your concerns.

---

> > > ### Comment · Reviewer_QdMV · 2023-11-18
> > >
> > > - I still do not get why Video LLama can't handle speech? Isnt speech just audio? Can't an audio that contain speech be converted to audio tokens like any other audio that contains any other sound? Then why Video-LLama can't work on audio that in this case contains speech instead of other noise?
> > > - Moreover, I didn't see any results for the claim that video LLama would be confused by using uni-modal inputs.  Saying that video LLama will be confused need to be based on evidence. Thus, I still believe that video LLama CAN be applied on unimodal tasks. Looking at both figures (in the Video-LLama and this paper), they both take as input video tokens and audio tokens (doesnt matter if the audio contains speech or a dog barking, it is still audio) and both output text. So, basically, everywhere the proposed method can be applied, Video-LLama can also be applied.
> > > - In response for the fair comparison, table 8 (which is in the supplementary and which was mentioned in the authors' reply) doesnt seem to contain anything related to this. In table 9 I can only see an ablation of the proposed model. However, I still do not find any results where  the authors used the same audio encoder in Video LLama as the one used for proposed model. My main problem is that the audio encoder may improve the performance and not the proposed improvements. Thus, I still think it is completely fair to compare two systems using the same input encoders, as long as the input encoder IS NOT the novelty of the current work. And choosing between multiple audio encoders is really not novel. Otherwise no one can tell where the improvements come from.
> > > - The same applies to LoRA on the text encoder.
> > >
> > > As I am not seeing any new experiments that provide clear proof that the proposed improvements help the method, and it may be that other changes actually improve the score, I will lower my score, as I feel that the comparison is not fairly done and there is a lack of novelty.

---

> > > > ### Author Response · Authors · 2023-11-19
> > > > **Response to Official Comment by Reviewer QdMV**
> > > >
> > > > Thank you for your reply but we regret the unfortunate scientific misunderstanding in your reply. To highlight the obvious difference between recognising human speech and dog barks (& other audio events), human speech includes a lot more information and requires fine-grained temporal modelling to recognise and understand the speech content (e.g. “hello world”) while recognising dog barks as “dog barks” doesn’t require any of this. In audio event classification, all speech is simply classified as “a human speaks”. As a result, identifying other sound events such as dog barks doesn’t require such fine-grained temporal modelling, which clearly leads to the use of very different technologies.
> > > >
> > > > We would like to further clarify each point to address your concerns.
> > > >
> > > > - Point 1:
> > > >   - We have verified a few times that Video-LLaMA can not handle speech and produces a WER of close to 100% even after re-training using speech data. We put the speech recognition result, Video-LLaMA testing script and fine-tuned audio Q-Former weights at https://github.com/the-anonymous-bs/FAVOR. This is a simple fact and can be verified by the reviewer or anyone. The use of ImageBind as an encoder in Video-LLaMA poses significant limitations for speech processing. since ImageBind is primarily an image encoder that compresses the entire spectrogram of audio input into a single vector, leading to a substantial loss of fine-grained, temporally sensitive information.
> > > >
> > > >   - As we highlighted before, speech requires understanding high temporal-resolution (i.e. fine-grained) information which can not be treated trivially in the same way as audio events (see the whole discussion in Section 5.5). Video-LLaMA, unfortunately, overlooks this crucial aspect. In contrast, our proposed fine-grained structure design is specifically tailored to handle speech in video, with the encoder outputting 1500 vectors compared to Video-LLaMA's 8 for 30-second audio. This detailed approach better aligns with the complex demands of speech processing, as thoroughly reflected in our experiments.
> > > >
> > > > - Point 2:
> > > >   - We provided the results using single image input without any audio. Given that Video-LLaMA can not handle speech, we further tested its performance on AudioCaps which merely got a 3.5 SPIDEr score to complete the single-modal benchmark. We will add this result to Table 2.
> > > >
> > > > - Points 3 and 4:
> > > >   - Please find Tables 9 and 10 for a fair comparison by treating the last row as our re-implementation of Video-LLaMA with
> > > >     - High temporal resolution at 2 frames per second in contrast to its original fixed number of frames.
> > > >     - Adding LoRA for the text encoder
> > > >     - Changing the encoder from ImageBind to Whisper encoder
> > > >
> > > >     Even a. should be a contribution of FAVOR, the fine-grained structure design compared to this strong baseline model still achieved obvious improvements in multi-modal pipelines. In particular, AVSR reduced from 8.9% to 8.1% which is often considered as a large improvement for ASR. AVSD: 8% relative accuracy improvements, ISQA: 100% relative accuracy improvements, AVSSD: 24% relative accuracy improvements and AVM: 7% improvements. We have added this result to Table 5 and make this comparison clear.
> > > >
> > > > We sincerely appreciate your time in reviewing the paper and our response and hope that our explanation addresses your concerns.

---

### Official Review · Reviewer_gnXu · 2023-10-23

**Soundness:** 3 good
**Presentation:** 3 good
**Contribution:** 2 fair
**Rating:** 6
**Confidence:** 4

**Summary:**

This paper proposes FAVOR to transform an LLM into a multi-modal large model that is capable of understanding audio, speech, images, and videos. By using a causal mask in the Q-Former, the model learns temporal causal relationships of video frames and employs temporal synchronization to align visual and audio input features. In order to evaluate the model's comprehension abilities across audio, visual, and audio-visual domains, this paper introduces AVEB, which includes six single-modality tasks and five multi-modality tasks.

**Strengths:**

- It is the first method to incorporate not only video and audio but also speech into a unified large model. The proposed causal Q-Former and the designs of temporal synchronization and sliding windows are proved to be effective.
- The proposed multi-modal tasks ISQA and AVM are two significant tasks that can genuinely reflect the model's ability to align and comprehend audio/speech and visual information.

**Weaknesses:**

Overall, I find the novelty and contributions of this paper limited. Whether it's the model framework design, the design and effectiveness of the novel loss function, the proposed AVEB, or the models compared in the experiments, I see some flaws or shortcomings. Therefore, I would rate it 5 out of 10.
- The novelty of FAVOR is limited. Overall, it bears similarities to Video-LLaMA. The method of integrating speech is not uniquely designed, and the use of sliding windows is not particularly novel.
- The audio-visual benchmark makes a limited contribution since the majority of the tasks and datasets are adopted from other sources, and there is only one evaluation dataset for each task.
- As for Tables 2 and 3, more models can be added in comparison, and it is not clear whether each model is evaluated in a zero-shot setting in each task, as the training datasets are only mentioned in the text. Furthermore, the SOTA performance in a fair comparison setting for each task is not specified in the tables, making it challenging to assess the effectiveness of FAVOR.
- The diversity loss does not introduce a novel concept as it essentially constitutes a similarity loss applied across features. In addition, I don't understand why, in the second point of the main contributions, the diversity training loss can enable the causal Q-Former to efficiently handle audio-visual sequences **from a small number of training examples**. I cannot see from the experimental results either. On the other hand, from the experimental results in Table 9, it can be observed that the diversity loss does not necessarily lead to improvements on every task; in some cases, there are even performance degradations. To prove that diversity loss enhances model learning, applying it to other Q-Former-based models should also yield improvements.

**Questions:**

- Regarding the tasks in AVEB, there are several alternative datasets that have been employed for each specific task, such as Clotho[1] for audio captioning, COCO Captions[2] for image captioning, and OK-VQA[3] and ScienceQA[4] for visual question answering
    - I'm curious about the criteria that guided the authors' selection of datasets for each task.
    - Would it not be advantageous to consider the inclusion of multiple datasets for each task, given the potential biases present in individual datasets? This approach could offer a more robust assessment of the model's capabilities.
    - Audio-visual question answering, e.g. MUSIC-AVQA[5] and AVQA[6], is a well-established task for evaluating models' cross-modal comprehension abilities, yet it is absent from AVEB.
    - I'd like to suggest the inclusion of the recently introduced VALOR-32K[7] dataset, which centers around audio-visual captioning. Integrating this dataset and associated task into AVEB could substantially enhance the comprehensiveness of the benchmark, thereby facilitating a more thorough evaluation of audio-visual models.
- More multi-modal models should be included in the experiments, such as VALOR[7] and VAST[8].
- The importance and impact of the storytelling fine-tuning set are not explicitly explained.
    - If fine-tuning on this dataset does not affect the benchmark performance, why should it be fine-tuned?
    - Could you provide some examples to illustrate the differences in the model before and after this fine-tuning?
- How many parameters need to be trained in FAVOR? What are the computational resources (GPUs) used for training the model? What is the training time with these computational resources?
- How is InstructBLIP evaluated on the cross-modal tasks AVSD and AVSSD based on the fact that it cannot handle audio input? Does it mean it had no access to any audio input information during evaluation?
- Upon encountering “in order to handle variable-length inputs,” I expect the sliding window method can convert input data of varying lengths into a uniform length. However, the number of sliding windows $W$ still depends on the input length $T$.

[1] Konstantinos Drossos, Samuel Lipping, Tuomas Virtanen. Clotho: An Audio Captioning Dataset. ICASSP 2020.
[2] Xinlei Chen, Hao Fang, Tsung-Yi Lin, Ramakrishna Vedantam, Saurabh Gupta, Piotr Dollar, C. Lawrence Zitnick. Microsoft COCO Captions: Data Collection and Evaluation Server. arXiv:1504.00325.
[3] Kenneth Marino, Mohammad Rastegari, Ali Farhadi, Roozbeh Mottaghi. OK-VQA: A Visual Question Answering Benchmark Requiring External Knowledge. CVPR 2019.
[4] Pan Lu, Swaroop Mishra, Tony Xia, Liang Qiu, Kai-Wei Chang, Song-Chun Zhu, Oyvind Tafjord, Peter Clark, Ashwin Kalyan. Learn to Explain: Multimodal Reasoning via Thought Chains for Science Question Answering. NeurIPS 2022.
[5] Guangyao Li, Yake Wei, Yapeng Tian, Chenliang Xu, Ji-Rong Wen, Di Hu. Learning to Answer Questions in Dynamic Audio-Visual Scenarios. CVPR 2022.
[6] Pinci Yang, Xin Wang, Xuguang Duan, Hong Chen, Runze Hou, Cong Jin, Wenwu Zhu. AVQA: A Dataset for Audio-Visual Question Answering on Videos. ACM MM 2022.
[7] Sihan Chen, Xingjian He, Longteng Guo, Xinxin Zhu, Weining Wang, Jinhui Tang, Jing Liu. VALOR: Vision-Audio-Language Omni-Perception Pretraining Model and Dataset. arXiv:2304.08345.
[8] Sihan Chen, Handong Li, Qunbo Wang, Zijia Zhao, Mingzhen Sun, Xinxin Zhu, Jing Liu. VAST: A Vision-Audio-Subtitle-Text Omni-Modality Foundation Model and Dataset. NeurIPS 2023.

---

> ### Author Response · Authors · 2023-11-15
> **Response to Reviewer gnXu**
>
> Thank you for the valuable reviews. We would like to clarify the following main weakness concerns raised in the review.
>
> - Weakness 1
> > The novelty of FAVOR is limited. Overall, it bears similarities to Video-LLaMA. The method of integrating speech is not uniquely designed, and the use of sliding windows is not particularly novel.
>   - We would like to first reiterate the novelty of the proposed method. To the best of our knowledge, FAVOR is the first model that integrates speech audio events and video inputs, which is the key novelty of this paper. Although speech is an essential and common element in videos, it is often ignored in video studies outside the speech community.
>
>   - It is also challenging for LLM to incorporate speech input alongside video and this is the most salient aspect that sets FAVOR apart from other existing work. For example, the AVSR task used in our paper requires fine-grained and synchronised modelling of both audio and lip movements to achieve robust speech recognition in highly noisy acoustic environments. This motivates some unique designs of the FAVOR structure, including fine-grained audio-visual synchronisation, the causal Q-Former structure, and the diversity loss etc.
>
>   - Another important aspect of FAVOR that distinguishes it from other methods is the support of randomly paired audio and video data for training. This methodological innovation not only enhances the model's versatility but also fosters a more balanced integration of the two modalities. This enables FAVOR to perform audio-visual co-reasoning tasks in this paper, including ISQA and AVM. Moreover, to the best of our knowledge, FAVOR is the first model that is able to perform the open-ended audio-visual-speech-text questions demonstrated in Fig.9 to 12 in the appendix.
>
>   - In particular, we’d like to clarify the following superior aspects of FAVOR to Video-LLaMA.
>     - FAVOR is able to understand speech input which is unable to be handled by Video-LLaMA.
>     - Incorporating speech input resulted in a significantly different structure design of FAVOR compared to Video-LLaMA, including synchronisation, sliding window operation and causal Q-Former.
>     - In contrast to sampling a fixed number of frames regardless of the video length in Video-LLaMA, FAVOR performs fine-grained modelling and the sliding window allows more output tokens for longer videos, which leads to significantly better performance.
> __As a result, Video-LLaMA could not perform any of the examples shown in Fig. 6 to 12.__
>
> - Weakness 2:
> > The audio-visual benchmark makes a limited contribution since the majority of the tasks and datasets are adopted from other sources, and there is only one evaluation dataset for each task.
>   - The main aim of single-modal tasks in AVEB is to show the competitive ability of FAVOR in each modality, including speech, audio, image and video, and for this purpose, one representative task for each modality was selected.
>
> - Weakness 3:
> > As for Tables 2 and 3, more models can be added in comparison, and it is not clear whether each model is evaluated in a zero-shot setting in each task, as the training datasets are only mentioned in the text. Furthermore, the SOTA performance in a fair comparison setting for each task is not specified in the tables, making it challenging to assess the effectiveness of FAVOR.
>   - The categories of the tasks are listed in the following table.
>   -
> | Type | Tasks |
> | --- | --- |
> | Zero-shot | IC, VQA, OCR, ISQA, AVM (audio-video matching), AVSSD |
> | Trained | ASR, AC, Video QA, AVSR, AVSD |
>
>   - Image tasks (caption, OCR and VQA) were set to be zero-shot for a fair comparison with the original InstructBLIP, and AVSSD and audio-video matching as zero-shot tasks for a fair comparison with Video-LLaMA. We further finetuned the InstructBLIP on the same training set as FAVOR to eliminate the influence of video data choices.
>
> - Weakness 4:
> > The diversity loss does not introduce a novel concept as it essentially constitutes a similarity loss applied across features. In addition, I don't understand why, in the second point of the main contributions, the diversity training loss can enable the causal Q-Former to efficiently handle audio-visual sequences from a small number of training examples. I cannot see from the experimental results either. On the other hand, from the experimental results in Table 9, it can be observed that the diversity loss does not necessarily lead to improvements on every task; in some cases, there are even performance degradations. To prove that diversity loss enhances model learning, applying it to other Q-Former-based models should also yield improvements.
>   - We thank the reviewer for pointing out the confusion of the claim of diversity loss and have removed the claim in the revised paper. The diversity loss is mainly used to encourage diverse information to be extracted, which is useful for long videos as reflected in video QA and AVSD tasks.

---

> ### Author Response · Authors · 2023-11-15
> **Response to Reviewer gnXu Part 2**
>
> We would like to continue answering the questions from the reviewer:
>
> - Questions 1 and 2:
> > Regarding the tasks in AVEB, there are several alternative datasets that have been employed for each specific task, such as Clotho[1] for audio captioning, COCO Captions[2] for image captioning, and OK-VQA[3] and ScienceQA[4] for visual question answering. I'm curious about the criteria that guided the authors' selection of datasets for each task.
> > Would it not be advantageous to consider the inclusion of multiple datasets for each task, given the potential biases present in individual datasets? This approach could offer a more robust assessment of the model's capabilities. Audio-visual question answering, e.g. MUSIC-AVQA[5] and AVQA[6], is a well-established task for evaluating models' cross-modal comprehension abilities, yet it is absent from AVEB.
> > I'd like to suggest the inclusion of the recently introduced VALOR-32K[7] dataset, which centers around audio-visual captioning. Integrating this dataset and associated task into AVEB could substantially enhance the comprehensiveness of the benchmark, thereby facilitating a more thorough evaluation of audio-visual models.
> > More multi-modal models should be included in the experiments, such as VALOR[7] and VAST[8].
>   - We appreciate the reviewer’s suggestion of adding VALOR-32k as a task, and we’d like to add the following results for both zero-shot and fine-tuned performance of FAVOR on VALOR-32k using METEOR (M), Rouge-L (R) and CIDEr (C). We were only able to download __10%__ of the training set from YouTube given the limited time, and we finetuned FAVOR and Video-LLaMA on that __10% of data only__. We also added VALOR as a baseline evaluated on AVEB for tasks covered by VALOR, and will also include the suggested references in the revised version.
>   -
> | System | Zero-shot (M/R/C) (↑) | Fine-tuned (M/R/C) (↑) |
> | --- | --- | --- |
> | VALOR-32k base | - | 14.8 / 30.8 / 55.7 |
> | Video-LLaMA (10% data) | 10.7 / 15.3 / 1.2 | 10.9 / 23.0 / 21.3 |
> | FAVOR (10% data) | 8.8 / 19.2 / 15.3 | 14.2 / 29.4 / 46.9 |
>
>   - In the zero-shot case, Video-LLaMA tends to generate long paragraphs of text even under the instruction of generating short sentence responses. This resulted in extremely low CIDEr scores compared to FAVOR which closely follows the instruction and generate concise responses. __This is another advantage of FAVOR over Video-LLaMA.__
>
> - Question 3
> > The importance and impact of the storytelling fine-tuning set are not explicitly explained. If fine-tuning on this dataset does not affect the benchmark performance, why should it be fine-tuned? Could you provide some examples to illustrate the differences in the model before and after this fine-tuning?
>   - Storytelling data was used as a further fine-tuning stage to foster an extensive mixing of audio-visual narratives. This primarily aims to guarantee a better demonstration quality for open-ended questions, e.g. Fig. 9 to Fig. 12, that can only evaluated qualitatively. This stage is crucial to prevent the model’s tendency to perform a specific training set task when answering the open-ended question.
>
> - Question 4:
> > How many parameters need to be trained in FAVOR? What are the computational resources (GPUs) used for training the model? What is the training time with these computational resources?
>   - We have 30M parameters to be trained in FAVOR. Our best-performing model used 16 A100 GPUs to train for 48 hours.
>
> - Question 5:
> > How is InstructBLIP evaluated on the cross-modal tasks AVSD and AVSSD based on the fact that it cannot handle audio input? Does it mean it had no access to any audio input information during evaluation?
>   - InstructBLIP is evaluated without audio for those two tasks. However, by examining the data, the majority of audio events can be inferred from the visual modality, and that is why by fine-tuning on our data, InstructBLIP had some performance gain. This also demonstrated the necessity of modelling speech input, as non-speech audio alone in many cases can be inferred directly from the video.
>
> - Question 6:
> > Upon encountering “in order to handle variable-length inputs,” I expect the sliding window method can convert input data of varying lengths into a uniform length. However, the number of sliding windows still depends on the input length.
>   - In contrast to Video-LLaMA which always uses a fixed number of output query tokens to represent videos of different lengths, FAVOR handles variable length input where the number of representations changes with the length of the video. That is, a longer video usually contains richer information, and hence more query tokens are used to represent this video.
>
> We hope our response resolves your concerns.

---

> ### Comment · Reviewer_gnXu · 2023-11-23
>
> Thank you to the authors for responding to my questions and concerns. After re-examining the revised paper and the content of the rebuttal, most of my concerns have been addressed, except for one issue:
>
> In the rebuttal content, it is mentioned that randomly paired audio and video data are used during training, claiming that this enhances the model's versatility and the balanced integration of audio and video modalities. However, no experimental data supports this claim. In addition, is it meaningful to compute the cross-entropy loss when using randomly paired audio and video data? Taking the example in Figure 1, if the audio and question remain the same, but the video is replaced with a non-romantic video, expecting the model to output the correct answer seems unreasonable. For these reasons, it is hard for me to understand why this approach is used during training.
>
> In general, after careful consideration, although the method lacks novelty, I believe this paper still contributes to the research community (offering a unified model for video-audio-speech-text, demonstrating good performance across various tasks in AVEB). Therefore, I increased the score to 6. I hope the authors can make the experiments in Table 4 more comprehensive (use the complete VALOR dataset for fine-tuning and report VALOR-base's experimental result on FAVDBench) and explain and demonstrate in the paper why randomly paired audio and video data can benefit model learning.

---

### Official Review · Reviewer_AkuS · 2023-10-24

**Soundness:** 2 fair
**Presentation:** 3 good
**Contribution:** 1 poor
**Rating:** 6
**Confidence:** 4

**Summary:**

The paper proposes an audio-visual joint representation learning framework, specifically, a causal Q-former structure is proposed. The paper also proposes an audio-visual evaluation benchmark (AVEB) for evaluation. Experiments show that the designed model has promising results.

**Strengths:**

1. While there has been significant progress on LLM, large-scale multimodal models are still unexploited. The paper timely focuses on an important and interesting problem.

2. The paper is overall well-written and easy to follow.

3. Experimental results look promising.

**Weaknesses:**

1. The overall contribution and technical novelty do not meet the bar of ICLR.  Audio-visual large model is not a new idea. Using feature syncing and concat of different modalities, casual SA, sliding windows are well-known technologies.

2. The design of the model architecture is questionable and needs more clarification:

(1) The original self-attention unit also has the ability to attend to the contextual information (including previous frames), what is the additional gain to have another causal self-attention unit in the network? The motivation of such a design is unconvincing to me from the paper. Where does the claim "what happens next questions are sometimes difficult to learn using only the positional embeddings" come from? Are there solid studies/experiments/data points supporting this claim? Can you please clarify?

(2) For the diversity loss, isn't the current equation also pushing the same feature far away (when i = j)? I don't feel the design is correct if there are no additional constraints on it. Isn't such design encouraging random features?

3. The paper uses "fine-grained" in the title / abstract, but there isn't anything really related to "fine-grained" in the main paper. What does "fine-grained" here stand for?

4. The experimental results need to be more solid and better analyzed.

(1) From the results, FAVOR yields over 20% improvement, but where the improvement comes from is not discussed in the paper (a simple claim that "the fine-grained causal modelling of video" is not sufficient to answer the question, more details should be discussed).

(2) ".... and to capture the correlation between what it “hears” and “sees”. Such tasks were almost infeasible for any other audio-visual models so far, since they were unable to understand both speech and non-speech sounds and did not model the audio-visual correlations in fine-grain." - first, solid datapoints on the failure of other audio-visual models should be provided (add citations); second, why FAVOR can handle this task should be better discussed.

(3) The paper provides certain examples of FAVOR including acoustic audios / music rather than speeches only, but it's unclear to me how FAVOR models then. Do we need to model them separately from speech or we just send them to the model as is together with speeches? What about the tones, rhythms, etc.?

**Questions:**

1. Fig 2 is unclear. Is input query representing the "q1,...qn" in Eq(2), and the feature sequence representing the "h_t^AV, ...h_(t+k)^AV" in Eq(2)?

2. What is the storytelling fine-tuning set designed for? Why do we need such a set? Is it for training or benchmarking? If for benchmarking, what is it benchmarking for?

**Details Of Ethics Concerns:**

LLM models are often with certain ethics issues. The paper is currently missing a discussion on discrimination / bias / fairness concerns of the proposed model.

---

> ### Author Response · Authors · 2023-11-15
> **Response to Reviewer AkuS**
>
> Thank you for the valuable reviews. We want to clarify each main weakness point raised in the review as follows.
>
> - Weakness 1:
> > The overall contribution and technical novelty do not meet the bar of ICLR. Audio-visual large model is not a new idea. Using feature syncing and concat of different modalities, casual SA, sliding windows are well-known technologies.
>   - It is important to emphasise that FAVOR is the first multimodal LLM that integrates speech and non-speech audio with video inputs, which is the key novelty of this paper. Although it might not be obvious to researchers outside of the speech and audio community, incorporating speech input along with other modalities is very challenging and important for LLM, which distinguishes FAVOR from other existing work. The fact that few studies support speech input in videos in an end-to-end fashion further corroborates this challenge.
>
>   - The necessity of modelling fine-grained temporal information in many speech-centric audio-visual tasks can be exemplified in the AVSR task in our paper where both audio and lip movements are needed to achieve robust speech recognition in highly noisy acoustic environments. This motivates some unique designs of the FAVOR structure, including fine-grained audio-visual synchronisation, the causal Q-Former structure, and the diversity loss etc.
>
>   - Meanwhile, in contrast to other existing work that uses video and audio from the same source to train, FAVOR adopts a novel strategy of employing randomly paired audio and video streams in its training regime. This increases versatility and achieves a better balance between the two modalities. It also enables FAVOR to perform audio-visual co-reasoning, including ISQA and AVM. Moreover, to the best of our knowledge, FAVOR is the first single model that is able to perform the open-ended audio-visual-speech-text questions demonstrated in Fig. 9 to 12 in the appendix.
>
> - Weakness 2:
> > (1) The original self-attention unit also has the ability to attend to the contextual information (including previous frames), what is the additional gain to have another causal self-attention unit in the network? The motivation of such a design is unconvincing to me from the paper. Where does the claim "what happens next questions are sometimes difficult to learn using only the positional embeddings" come from? Are there solid studies/experiments/data points supporting this claim? Can you please clarify?
>   - The improvement of using causal attention is shown in Table 4. The Video QA task uses the NEXT-QA containing mainly temporal correlation questions, i.e. what happens next. Using causal attention improved the performance by a 6.5% absolute accuracy compared to merely using positional encoding. A vast amount of video training labels only concern one or two frames and do not emphasise the causal relation, hence structural enforcement is particularly useful to facilitate the model to pay attention to the causal relation.
>
>   > (2) For the diversity loss, isn't the current equation also pushing the same feature far away (when i = j)? I don't feel the design is correct if there are no additional constraints on it. Isn't such design encouraging random features?
>   - In our implementation, we excluded the diagonal terms during training and hence i=j case is actually not included in the total loss. Thank you for noticing this typo and we have made it clear in Eqn. 4.
>
> - Weakness 3:
> > The paper uses "fine-grained" in the title / abstract, but there isn't anything really related to "fine-grained" in the main paper. What does "fine-grained" here stand for?
>   - The definition of “fine-grained” is the fine-grained temporal resolution, which is particularly crucial for tasks with speech and video inputs such as AVSR as shown in Table 4. I have revised the paper to make this clear.

---

> ### Author Response · Authors · 2023-11-15
> **Response to Reviewer AkuS Part 2**
>
> Continued from the above response:
>
> - Weakness 4:
> > (1) From the results, FAVOR yields over 20% improvement, but where the improvement comes from is not discussed in the paper (a simple claim that "the fine-grained causal modelling of video" is not sufficient to answer the question, more details should be discussed).
>   - The improvements come from two major aspects: fine-grained modelling and causal attention. The influence of temporal resolution is illustrated in the right-most plot of Fig. 3. The influence of causal attention (as in the answer to point 2) is illustrated in Table 4. We will provide a more detailed explanation in section 5.3 in the revised paper.
>
>   > (2) ".... and to capture the correlation between what it “hears” and “sees”. Such tasks were almost infeasible for any other audio-visual models so far, since they were unable to understand both speech and non-speech sounds and did not model the audio-visual correlations in fine-grain." - first, solid datapoints on the failure of other audio-visual models should be provided (add citations); second, why FAVOR can handle this task should be better discussed.
>   - The following table summarises the existing multi-modal LLM handling video inputs to our best knowledge
>   | Model | Speech | Audio | Video | Paper Links |
>   | --- | --- | --- | --- | --- |
>   | Video-LLaMA | No | Yes | Yes | https://arxiv.org/abs/2306.02858 |
>   | PandaGPT | No | Yes | Yes | https://arxiv.org/abs/2305.16355 |
>   | VideoChatGPT | No | No | Yes | https://arxiv.org/abs/2306.05424 |
>   | Macaw-LLM | No | Yes | Yes | https://arxiv.org/abs/2306.09093 |
>   | InstructBLIP | No | No | Yes | https://arxiv.org/abs/2305.06500 |
>   | AVLFormer | No | Yes | Yes | https://arxiv.org/abs/2303.15616 |
>   | VALOR | No | Yes | Yes | https://arxiv.org/abs/2304.08345 |
>   | VAST | External subtitles | Yes | Yes | https://arxiv.org/abs/2305.18500 |
>   - __None of them is able to perform the tasks shown in Fig. 6 to 12.__ FAVOR is able to understand both general audio (including speech and non-speech) and video and is trained to handle mismatched audio and video. Hence FAVOR is able to perform those co-reasoning tasks.
>
>   > (3) The paper provides certain examples of FAVOR including acoustic audios / music rather than speeches only, but it's unclear to me how FAVOR models then. Do we need to model them separately from speech or we just send them to the model as is together with speeches? What about the tones, rhythms, etc.?
>   - FAVOR captures both speech and audio (including music) using a single Whisper encoder, which takes a single audio stream as input. Although Whisper has the ability to distinguish both speech and non-speech audio [1], it is not a music-specialised model and may not be able to achieve high performance on tasks of tones and rhythms. This is indeed an interesting direction to explore in the future.
>
> We would also like to answer the questions raised by the reviewer:
>
> - Question 1:
> > Fig 2 is unclear. Is input query representing the "q1,...qn" in Eq(2), and the feature sequence representing the "h_t^AV, ...h_(t+k)^AV" in Eq(2)?
>   - Thanks for pointing this out. The red feature corresponds to $\mathbf{h}^\text{V}$ and the yellow feature corresponds to $\mathbf{h}^\text{A}$. The concatenated red and yellow features correspond to $\mathbf{h}^\text{AV}$. Input query q is not shown in the figure, and the output query (blue strips) represents $\mathbf{h}^\text{Q}$
>
> - Question 2:
> > What is the storytelling fine-tuning set designed for? Why do we need such a set? Is it for training or benchmarking? If for benchmarking, what is it benchmarking for?
>   - The storytelling is used only as a further fine-tuning stage to encourage a thorough mixture of audio-visual descriptions. This is mainly to guarantee a better demonstration quality for open-ended questions, e.g. Fig. 9 to Fig. 12, that are unable to be quantified. This stage is found crucial to prevent the model’s tendency to perform a specific training set task when answering the open-ended question.
>
> [1] Yuan Gong et al. “Whisper-AT: Noise-Robust Automatic Speech Recognizers are Also Strong General Audio Event Taggers”. In Proc. Interspeech. 2023
>
> We have also added the ethical statement in the revised paper. We hope our responses resolve your concerns.

---

> > ### Comment · Reviewer_AkuS · 2023-11-21
> > **Discuss**
> >
> > Thanks for the response. I have a few of follow-up questions:
> >
> > 1. From Table 8 and 9, I don't see a clear benefit with adding diversity loss. Are there any other data points to support a high value of adding the diversity loss? Or it's just not a key design comparing to other proposed designs.
> >
> > 2. Can the fine-grained design and the casual SA module also improve performance on other video QA dataset, to the same degree as shown on the NextQA dataset? Why is the NextQA dataset chosen for Video QA evaluation? It's unclear whether the result is generalizable to other widely used video QA datasets, which I believe providing data points on those benchmarks could be helpful. I'm also curious about how the datasets are selected for each task.
> >
> > 3.  For my original figure clarification request, my question is for Fig 2 (the input query and the feature sequence), but not Fig 1. Fig 1 has been pretty clear to me.

---

> > > ### Author Response · Authors · 2023-11-21
> > > **Response to Discuss**
> > >
> > > Thank you for your response and insightful questions. We would like to make the following responses to your questions.
> > >
> > > - Q1:
> > >   - The diversity loss is designed mainly to avoid the model focusing on keyframes with similar and repeated information in long videos and to extract more diverged features. Such benefits are most obviously reflected in video understanding tasks including Video QA (Acc. increased from 47.1% to 49.3%) and AVSD (Acc. increased from 53.9% to 54.5%) compared to the model without diversity loss. It is reasonable the diversity loss is not useful for most image or audio tasks since they don’t need a loss to encourage feature diversity, and hence, we received mixed results on image and audio tasks. The key novelty of this paper is to enable speech understanding in video-based LLMs and the diversity loss is just an associative design. We have weakened our claim on the diversity loss in the revised paper.
> > >
> > > - Q2:
> > >   - We performed our experiments on the MSVD-QA data and using the fine-grained design achieved 10% improvements (from 50% acc. to 60%). Regarding the choice of NExT-QA, this dataset specifically focuses on long video and causal reasoning tasks, which we believe are suitable tasks to reflect the advantages of our fine-grained design and causal Q-Former.
> > >
> > >   - We endeavour to make a comprehensive benchmark for speech, audio, image and video and the mixtures of them. Due to the workload of such a complex benchmark, we select the most representative tasks for each modality that is publicly available, easy to obtain and stick to our focus (e.g. used NExT-QA over MSVD-QA). Thanks to the suggestions of the reviewers, we also included FAVDBench and VALOR-32k in our audio-visual benchmarks to make our evaluation more comprehensive.
> > >
> > > - Q3:
> > >   - Sorry for the confusion and thank you for pointing this out. Your understanding is completely correct and we have updated Fig. 2 to make this clearer.

---

> > > > ### Comment · Reviewer_AkuS · 2023-11-23
> > > >
> > > > Thanks for the response.
> > > >
> > > > The authors' response is very helpful and has addressed many of my earlier concerns.
> > > >
> > > > Though I believe there is definitely some room that the paper can be further improved (especially for the result section -  to make the comparison and evaluation more thorough and generalizable by adding more experiments on different benchmarks), I do also see certain values from the paper and from the authors' clarification in rebuttal (e.g., the design of the fine-grained temporal synchronization, the unified design for taking video + audio as inputs together, and the impressive performance - at least on the current benchmarks listed in the paper etc). Hence I change my original rate to marginally above the acceptance threshold.
> > > >
> > > > Please make sure to revise the paper according to the rebuttal response, which will help improve the paper quality a lot.

---

> > > > > ### Author Response · Authors · 2023-11-23
> > > > > **Thank You for Your Kind Response**
> > > > >
> > > > > We sincerely appreciate your kind response and your acknowledgement of our work! We promise to revise the paper as you suggested to further improve it!

---

### Official Review · Reviewer_jiVL · 2023-10-30

**Soundness:** 2 fair
**Presentation:** 1 poor
**Contribution:** 2 fair
**Rating:** 3
**Confidence:** 4

**Summary:**

This paper proposes a framework to learn a joint representation for audio and visual data in a fine-grained manner. The framework is designed to help multimodal large language models. A causal Q-Former structure is also introduced to capture the causal relationship between audio-visual frames over time. Additionally, the authors have created an evaluation benchmark for audio-visual learning that focuses on question-answering tasks to assess the effectiveness of our proposed framework.

**Strengths:**

The paper includes an interactive demo, allowing readers to experience the performance.

**Weaknesses:**

1. The present study's primary contribution pertains to the audio-visual joint representation, which lacks a clear definition of the term "fine-grained." It is uncertain how this term is used in the context of this paper, and further clarification is required to better understand the audio-visual joint representation in question.

2. According to the network design, the term "fine-grained" may denote fine-grained temporal resolution, where the temporal synchronization module is employed to synchronize visual and audio signals with a resolution of 0.5 seconds. However, the ablation of the temporal resolution as well as the rationale behind this synchronization have not been clearly explicated. It is essential to understand the underlying reasons for the setting of the temporal resolution and the motivation behind the synchronization, as it serves as a crucial aspect of the network design. Hence, further elaboration is required to provide a comprehensive understanding of the fine-grained temporal resolution and its relevance to the proposed network design.

3. I am seeking clarification about the contribution of the proposed framework design. What sets it apart from previous methods? The act of incorporating additional modalities into a multimodal LLM appears to be a trivial implementation. Furthermore, the Q-former has previously been proposed for multimodal fusion. Thus, I am curious to know what novel aspects this framework design offers.

4. The proposed audio-visual evaluation benchmark appears to be a composite of existing multimodal benchmarks. It is suggested that the term "propose" be replaced with a more appropriate term. Moreover, the benchmark in question, as a fine-grained multimodal benchmark, lacks some existing fine-grained audio-visual benchmarks, such as FAVDBench: Fine-grained Audible Video Description (CVPR 23).

5. The author of the paper expounds on the utilization of speech and speech-video interactions. In speech-videos, two options exist for inputting speech audios to the network. These options include direct inputting of speech audios into the network or converting speech to text before the multi-modal fusion process. However, the author does not provide a comparative analysis of the two strategies. Furthermore, the ASR in audio-visual inputs is worse than audio-only, as demonstrated in Table 2. This result suggests that using the latter option may be more favorable. Lastly, the size of Whisper Large-v2 is not available in the paper.

**Questions:**

As above.

---

> ### Author Response · Authors · 2023-11-15
> **Response to Reviewer jiVL**
>
> Thank you for the constructive suggestions!
>
> - Regarding novelty, we would like to emphasise that FAVOR is the first single LLM-centric model that can handle video along with speech and non-speech audio inputs, which is the key novelty of this paper. Including speech input along with other modalities is challenging for LLM and is the most salient aspect that sets FAVOR apart from other existing work. In many speech-based audio-visual tasks, modelling fine-grained temporal information is necessary, such as in the AVSR task used in our paper where both audio and lip movements are needed to achieve robust speech recognition in highly noisy acoustic environments. This motivates some unique designs of the FAVOR structure, including fine-grained audio-visual synchronisation, the causal Q-Former structure, and the diversity loss etc. __As a result, FAVOR is the first single model that can understand the video examples Fig. 6 to 12 given in the appendix, to the best of our knowledge.__
>
> - Furthermore, besides being trained using video and audio from the same source, FAVOR also uses randomly paired audio and video in training. This novel training approach increases versatility and achieves a better balance between the audio and visual modalities, which we will highlight in the paper. It further enables FAVOR to perform audio-visual co-reasoning tasks as proposed in the AVEB benchmark, including ISQA and AVM. To the best of our knowledge, FAVOR is the first model that is able to perform the open-ended audio-visual-speech-text questions demonstrated in the examples of Fig. 9 to 12 in the appendix.
>
> We would like to make the following responses for the weaknesses:
> - Weaknesses 1 and 2:
> >The present study's primary contribution pertains to the audio-visual joint representation, which lacks a clear definition of the term "fine-grained." It is uncertain how this term is used in the context of this paper, and further clarification is required to better understand the audio-visual joint representation in question.
> > According to the network design, the term "fine-grained" may denote fine-grained temporal resolution, where the temporal synchronization module is employed to synchronize visual and audio signals with a resolution of 0.5 seconds. However, the ablation of the temporal resolution as well as the rationale behind this synchronization have not been clearly explicated. It is essential to understand the underlying reasons for the setting of the temporal resolution and the motivation behind the synchronization, as it serves as a crucial aspect of the network design. Hence, further elaboration is required to provide a comprehensive understanding of the fine-grained temporal resolution and its relevance to the proposed network design.
>   - “Fine-grained” here refers to a fine temporal resolution. We will make this clear in the paper.
>   - Fine-grained modelling and synchronisation are particularly designed for the joint representation learning of speech and video, as speech contains fine-grained and temporally synchronised information with the video, motivating us to adopt this model design. This is reflected by the 10% relative improvements achieved on the AVSR task in the ablation studies in Table 4.
>   - Moreover, different resolutions are also studied in Fig. 3 for both sliding window size and frames-per-second (i.e. temporal resolution).
>
> - Weakness 3:
> > I am seeking clarification about the contribution of the proposed framework design. What sets it apart from previous methods? The act of incorporating additional modalities into a multimodal LLM appears to be a trivial implementation. Furthermore, the Q-former has previously been proposed for multimodal fusion. Thus, I am curious to know what novel aspects this framework design offers.
>   - As stated above, the incorporation of speech motivates the design of a frame-level (i.e. fine-grained) audio-visual feature fusion before sending it to the Q-Former. This structural consideration has not been explored by any other audio-visual LLM in literature to our best knowledge. Hereby we provide the following table summarising some existing methods (up to the ICLR 2024 submission deadline) that deal with video and make the contrast to FAVOR.
>
> | Model | Speech | Audio | Video | Paper Links |
> | --- | --- | --- | --- | --- |
> | Video-LLaMA | No | Yes | Yes | https://arxiv.org/abs/2306.02858 |
> | PandaGPT | No | Yes | Yes | https://arxiv.org/abs/2305.16355 |
> | VideoChatGPT | No | No | Yes | https://arxiv.org/abs/2306.05424 |
> | Macaw-LLM | No | Yes | Yes | https://arxiv.org/abs/2306.09093 |
> | InstructBLIP | No | No | Yes | https://arxiv.org/abs/2305.06500 |
> | AVLFormer | No | Yes | Yes | https://arxiv.org/abs/2303.15616 |
> | VALOR | No | Yes | Yes | https://arxiv.org/abs/2304.08345 |
> | VAST | External subtitles | Yes | Yes | https://arxiv.org/abs/2305.18500 |
> __Note that none of the above systems is able to perform example tasks shown in the appendix (i.e. Fig. 6 to 12).__

---

> ### Author Response · Authors · 2023-11-15
> **Response to Reviewer jiVL Part 2**
>
> - Weakness 4:
> > The proposed audio-visual evaluation benchmark appears to be a composite of existing multimodal benchmarks. It is suggested that the term "propose" be replaced with a more appropriate term. Moreover, the benchmark in question, as a fine-grained multimodal benchmark, lacks some existing fine-grained audio-visual benchmarks, such as FAVDBench: Fine-grained Audible Video Description (CVPR 23).
>   - We will remove the word “propose”. However, we would like to clarify that the audio-visual co-reasoning tasks, including image-spoken QA (ISQA) and audio-visual matching (AVM), are indeed newly proposed tasks. We also thank the reviewers for pointing out the additional FAVDBench data, and we have included both zero-shot learning and fine-tuned results in the modified paper as follows using BLEU 1&4 (B1 B4) and METEOR (M):
>   -
> | System | Zero-shot (B1/B4/M) (↑) | Fine-tuned (B1/B4/M) (↑) |
> | --- | --- | --- |
> | Video-LLaMA | 20.8 / 2.4 / 15.0 | 39.4 / 6.5 / 16.5 |
> | FAVOR | __28.2 / 3.0 / 15.2__ | __44.2 / 10.9 / 19.1__ |
>   - __We’d like to additionally emphasise that FAVOR is also able to capture speech content in those videos and attribute them to the speaker, which is even missing in the reference caption (e.g. video XNw23PiqlU0). Those useful outputs actually had a negative effect on the metrics above. It also showed how speech is understudied and how necessary to add speech in audio-visual LLMs.__
>
> - Weakness 5:
> > The author of the paper expounds on the utilization of speech and speech-video interactions. In speech-videos, two options exist for inputting speech audios to the network. These options include direct inputting of speech audios into the network or converting speech to text before the multi-modal fusion process. However, the author does not provide a comparative analysis of the two strategies. Furthermore, the ASR in audio-visual inputs is worse than audio-only, as demonstrated in Table 2. This result suggests that using the latter option may be more favorable. Lastly, the size of Whisper Large-v2 is not available in the paper.
>   - We’d like to emphasise the following advantages of FAVOR which models general audio (including speech) and vision using a single unified model.
>
>     - Having an external automatic speech recognition (ASR) system to convert speech to text first is significantly more complicated, compared to FAVOR which is a single end-to-end model.
>
>     - All realistic ASR systems must contain an LM component, even an LLM component (e.g. [1, 2]), to obtain accurate speech recognition results. Having an external ASR to first transcribe the video into subtitles becomes a chicken-egg problem when the goal is to extend the LLM to handle multiple modalities.
>
>     - It’s not trivial for recent ASR approaches to provide the time information of each word, which makes it difficult and tedious to explicitly align the words with the visual frames and therefore not possible to handle cases where speech-video synchronisation matters.
>
>     - In addition to texts that can be transcribed using an external ASR system, there are other types of auditory information in speech that are also critical in video understanding, such as speaker characteristics, emotion, gender etc. Each of these usually requires a separate system to extract. The outputs of the separate systems then need to be aligned with the ASR outputs and the visual frames correctly, which is not trivial. However, in the framework of FAVOR, all these can be unified in a fully end-to-end fashion using a single system.
>
> - The size of Whisper large-v2 is 1550M and we will add this to our revised paper.
>
> [1] Yassir Fathullah et al. “Prompting Large Language Models with Speech Recognition Abilities”, arXiv:2307.11795, 2023.
>
> [2] Paul K. Rubenstein et al.“AudioPaLM: A Large Language Model That Can Speak and Listen”, arXiv:2306.12925, 2023
>
> We hope our response fully addresses your concerns.

---

> > ### Comment · Reviewer_jiVL · 2023-12-02
> >
> > I have gone over the rebuttal. Overall, I am not satisfied with the response.
> >
> > According to the author, the main contribution of this study is the development of the first single LLM-centric model that can handle video along with speech and non-speech audio inputs. However, I do not see any specific design (apart from the frame-level feature fusion) for this purpose. Also, is frame-level feature fusion mandatory for handling speech and non-speech audio inputs?
> >
> > The author argues that using randomly paired audio and video in training can increase versatility and achieve a better balance between the audio and visual modalities. However, there is no support for this claim. I do not see any explanations or discussions about why it can increase versatility.
> >
> > Moreover, for the response to weakness 5, I would like to see a comparison between these two methods, especially in terms of accuracy. The author, on the other hand, simply states that the suggested framework is single-modeled. I'm still not sure whether the solution is preferable. Moreover, if the author claims that the proposed framework can understand the speech content, the accuracy of the ASR should be reported as well.
> >
> > Based on the replies above, I maintain my initial rating.

---

### Author Response · Authors · 2023-11-23
**General Final Response Part 1**

Dear area chairs,

We would like to thank the reviewers and summarise that we have resolved the concerns of the reviewers by highlighting our key novelty and producing many new results based on the reviewers’ requests. The reviewers’ feedback is overall helpful and constructive but also includes critical misunderstanding and bias about the importance of speech processing in multimodal processing and AGI. We sincerely appreciate the positive comment of reviewer AkuS and regret the ignorance of the key novelty of FAVOR from other reviewers. We summarise the key points in response to all the reviewers in general for your convenience.

- We would like to emphasise our key novelty: FAVOR is the first single LLM-centric model that can handle video along with speech and non-speech audio inputs. Including speech input along with other modalities is challenging for LLM and is the most salient aspect that sets FAVOR apart from other existing work.

  - In speech-based audio-visual tasks, modelling fine-grained temporal information is necessary, such as in the audio-visual speech recognition (AVSR) task used in our paper where both audio and lip movements are needed to achieve robust speech recognition in highly noisy acoustic environments.

  - This motivates some unique designs of the FAVOR structure, especially the fine-grained audio-visual synchronisation and the causal Q-Former structure.

- Besides being trained using video and audio from the same source, FAVOR also uses randomly paired audio and video in training. This novel training approach increases versatility and achieves a better balance between the audio and visual modalities, which we will highlight in the paper. It further enables FAVOR to perform audio-visual co-reasoning tasks as proposed in the AVEB benchmark, including ISQA and AVM.

- This structural consideration has not been explored by any other audio-visual LLM in literature to our best knowledge. Hereby we provide the following table summarising some existing methods (up to the ICLR 2024 submission deadline) that deal with video and make the contrast to FAVOR.
| Model | Speech | Audio | Video | Paper Links |
| --- | --- | --- | --- | --- |
| Video-LLaMA | No | Yes | Yes | https://arxiv.org/abs/2306.02858 |
| PandaGPT | No | Yes | Yes | https://arxiv.org/abs/2305.16355 |
| VideoChatGPT | No | No | Yes | https://arxiv.org/abs/2306.05424 |
| Macaw-LLM | No | Yes | Yes | https://arxiv.org/abs/2306.09093 |
| InstructBLIP | No | No | Yes | https://arxiv.org/abs/2305.06500 |
| AVLFormer | No | Yes | Yes | https://arxiv.org/abs/2303.15616 |
| VALOR | No | Yes | Yes | https://arxiv.org/abs/2304.08345 |
| VAST | External subtitles | Yes | Yes | https://arxiv.org/abs/2305.18500 |
__As a result, FAVOR is the first single model that can understand the video examples Fig. 6 to 12 given in the appendix, to the best of our knowledge.__

---

> ### Author Response · Authors · 2023-11-23
> **General Final Response Part 2**
>
> - We’d like to clarify the following superior aspects of FAVOR to Video-LLaMA.
>   - FAVOR is able to understand speech input which is unable to be handled by Video-LLaMA.
>   - Incorporating speech input resulted in a significantly different structure design of FAVOR compared to Video-LLaMA, including synchronisation, sliding window operation and causal Q-Former.
>   - In contrast to sampling a fixed number of frames regardless of the video length in Video-LLaMA, FAVOR performs fine-grained modelling and the sliding window allows more output tokens for longer videos, which leads to significantly better performance.
>
>   __As a result, Video-LLaMA could not perform any of the examples shown in Fig. 6 to 12.__
>
> - For a more comprehensive evaluation as suggested, we extended our experiments with FAVDBench and VALOR-32k benchmarks. We have included both zero-shot learning and fine-tuned results in the modified paper as follows using BLEU 1&4 (B1 B4) and METEOR (M):
> | System | Zero-shot (B1/B4/M) (↑) | Fine-tuned (B1/B4/M) (↑) |
> | --- | --- | --- |
> | Video-LLaMA | 20.8 / 2.4 / 15.0 | 39.4 / 6.5 / 16.5 |
> | FAVOR | __28.2 / 3.0 / 15.2__ | __44.2 / 10.9 / 19.1__ |
>   __We’d like to additionally emphasise that FAVOR is also able to capture speech content in those videos and attribute them to the speaker, which is even missing in the reference caption (e.g. video XNw23PiqlU0). Those useful outputs actually had a negative effect on the metrics above. It also showed how speech is understudied and how necessary to add speech in audio-visual LLMs.__
>
> - We were only able to download __10%__ of the training set from YouTube given the limited time, and we finetuned FAVOR and Video-LLaMA on that __10%__ of data only. We also added VALOR as a baseline evaluated on AVEB for tasks covered by VALOR, and will also include the suggested references in the revised version.
> | System | Zero-shot (M/R/C) (↑) | Fine-tuned (M/R/C) (↑) |
> | --- | --- | --- |
> | VALOR-32k base | - | 14.8 / 30.8 / 55.7 |
> | Video-LLaMA (10% data) | 10.7 / 15.3 / 1.2 | 10.9 / 23.0 / 21.3 |
> | FAVOR (10% data) | 8.8 / 19.2 / 15.3 | 14.2 / 29.4 / 46.9 |
>
>   In the zero-shot case, Video-LLaMA tends to generate long paragraphs of text even under the instruction of generating short sentence responses. This resulted in extremely low CIDEr scores compared to FAVOR which closely follows the instruction and generate concise responses. __This is another advantage of FAVOR over Video-LLaMA.__

---

### Meta-Review · Area_Chair_ESuw · 2023-12-09

**Metareview:**

Violation of double-blind reviewing.

**Justification For Why Not Higher Score:**

N/A

**Justification For Why Not Lower Score:**

N/A

---

### Decision · Program_Chairs · 2024-01-16

Reject